# Embryonic macrophages orchestrate niche cell homeostasis for the establishment of the definitive hematopoietic stem cell pool

Gülce Perçin [1] ✉, Konstantin Riege [2], Julia Fröbel [1], Jonas Metz [3], Stephan Culemann [1], Mathias Lesche [4], Susanne Reinhardt [4], Thomas Höfer [3], Steve Hoffmann [2,5] & Claudia Waskow [1,5,6] ✉

Embryonic macrophages emerge before the onset of definitive hematopoiesis, seed into discrete tissues and contribute to specialized resident macrophages throughout life. Presence of embryonic macrophages in the bone marrow and functional impact on hematopoietic stem cells (HSC) or the niche remains unclear. Here we show that bone marrow macrophages consist of two ontogenetically distinct cell populations from embryonic and adult origin. Newborn mice lacking embryonic macrophages have decreased HSC numbers in the bone marrow suggesting an important function for embryo-derived macrophages in orchestrating HSC trafficking around birth. The establishment of a normal cellular niche space in the bone marrow critically depends on embryonic macrophages that are important for the development of mesenchymal stromal cells, but not other non-hematopoietic niche cells, providing evidence for a specific role for embryo-derived macrophages in the establishment of the niche environment pivotal for the establishment of a normally sized HSC pool.

Definitive hematopoietic stem cells (HSC) are the key reservoir for lifelong blood cell formation and their function dynamically adjusts to the demands at different ages and is regulated -at least in part- through interaction with a functional stem cell niche[1–3]. HSCs arise during embryonic development and migrate to the bone marrow space around birth[4,5] and are sensitive to inflammatory stimuli during this time period[6,7]. However, the historically predominant concept of HSCs expanding during development within the fetal liver, followed by subsequent migration to the bone marrow has recently been challenged. Instead, HSCs show limited proliferation in the fetal liver[8] but rather expand postnatally after transfer to the bone marrow niche space that only then becomes supportive to host definitive HSCs[9]. These insights emphasize the importance of characterizing regulatory mechanisms responsible for the establishment of a functional stem cell niche in the bone marrow, which may also lead to novel approaches for the expansion of functional HSCs in vitro.

In the adult mouse, many different non-hematopoietic, bone marrow-resident cell types have been implicated in the regulation of HSC function (summarized in ref. 2,10). However, regulatory mechanisms responsible for the setup of the stem cell niche and factors involved in recruiting HSCs during development remain incompletely understood. Mesenchymal stromal cells (MSCs) and endothelial cells (EC) are important niche cells and form a microenvironment supporting definitive HSC pool establishment. Specifically, neural crest-derived mesenchymal cells progressively constitute the bone marrow stem cell niche around birth and are important for

[1]Immunology of Aging, Leibniz Institute on Aging - Fritz Lipmann Institute (FLI), Jena, Germany. [2]Computational Biology of Aging, Leibniz Institute on Aging - Fritz Lipmann Institute (FLI), Jena, Germany. [3]Theoretical Systems Biology, German Cancer Research Center, Heidelberg, Germany. [4]DRESDEN-concept Genome Center, c/o CMCB Center for Molecular and Cellular Bioengineering Technology Platform of the TUD Dresden University of Technology, Dresden, Germany. [5]Institute of Biochemistry and Biophysics, Faculty of Biological Sciences, Friedrich-Schiller-University, Jena, Germany. [6]Department of Medicine III, Faculty of Medicine, TU Dresden, Dresden, Germany. ✉e-mail: Guelce.Percin@leibniz-fli.de; Claudia.waskow@leibniz-fli.de

the recruitment of HSCs via the production of CXCL12[11]. These niche cells may be overlapping with *Prx*-positive cells expressing the transcription factor Foxc1 that regulates the production of CXCL12[12]. CXCL12 produced by *Lepr*-expressing niche cells[13], as well as SCF produced by endothelial cells, proved also important for the proper establishment of a definitive HSC pool in neonate mice[14]. In this context, information on the cellular nature of physiological regulators orchestrating the establishment of a functional stem cell niche, niche cell composition, and their cellular interplay in the bone marrow is missing.

Tissue-resident macrophages (TR-Mp) emerge in the yolk sac during development earlier than HSCs[15], seed many different tissues and are largely maintained throughout lifelong self-renewal in situ (reviewed in refs. [16–18]). TR-Mps include microglia in the brain, Kupffer cells in the liver, alveolar macrophages in the lung, Langerhans cells in the skin, red pulp macrophages in the spleen, fat-associated macrophages, Lamina Propria macrophages in the gut, and large peritoneal macrophages. In the bone marrow, only highly specialized macrophages, the osteoclasts, are so far considered to be of embryonic origin[17–19]. Functionally, TR-Mps integrate cues on tissue dysfunction and infection and provide factors important for tissue growth, homeostasis, and repair to tightly associated tissue-specific cells. Besides a direct role in the sensing of danger signals, embryonic TR-Mps also regulate the establishment and homeostasis of key immune modulatory cells, the dendritic cells in the spleen[20], and are important for iron homeostasis[21,22], providing evidence for the role of an embryo-derived cell type in establishing and maintaining the tissue-specific structural integrity of this organ. Also, microglia in the brain control structural integrity between fetal cortical boundaries[23], and peritoneal macrophages control immunoglobulin production by peritoneal B1 cells[24], suggesting a tissue-overarching function of embryonic macrophages. However, a unifying understanding of how embryonic tissue-resident macrophages in the bone marrow relate to other niche cells and regulate hematopoiesis remains elusive.

Here we use tissue-specific lineage tracing and deletion of key signals pivotal for the development of embryonic macrophages to dissect their role in the bone marrow in vivo. Thereby we define important functions for bone marrow-resident embryonic macrophages in the set-up of a functional stem cell niche and the establishment of a normal definitive HSC pool in situ.

## Results

### Loss of HSCs in young mice devoid of embryonic myeloid cells

To test for a role of embryonic versus adult myeloid cells in HSC generation, *Csf1r*[−/−], *Rank*[cre/+];*Csf1r*[fl/−], and *Vav*[cre/+];*Csf1r*[fl/−] mice were generated. Embryonic, but not adult HSC-derived macrophages require Csf1 receptor signaling for their generation[20]. Rank-cre specifically induces recombination of LoxP-flanked alleles in embryonic macrophages but not in other embryonic or adult-derived immune cells[19,20] and, as a consequence, *Rank*[cre/+];*Csf1r*[fl/−] mice lack embryonic macrophages including osteoclasts and are osteopetrotic[19]. Hematopoietic progenitors (HPC, Kit[+] Sca-1[+] Lin[−], KSL), stem cells (HSC, KSL CD48[−] CD150[+], KSL Slam) and common lymphoid progenitors (CLP), but not myeloid hematopoietic progenitor populations (CMP, GMP, MEP) are reduced in *Csf1r*[−/−] and *Rank*[cre/+];*Csf1r*[fl/−] but not in *Vav*[cre/+];*Csf1r*[fl/−] mice; the latter express CRE recombinase in all definitive HSCs and their cellular progeny (Fig. 1a, b, Supplementary Fig. 1a). There is no difference in HSC phenotype between *Rank*[cre/+];*Csf1r*[fl/−] and *Rank*[cre/+];*Csf1r*[fl/fl] mice. Due to reduced bone marrow cavity space (bone marrow volume)[19], total leukocyte numbers are decreased in *Csf1r*[−/−] and *Rank*[cre/+];*Csf1r*[fl/−] mice but the reduction of HSPCs outnumbers this overall reduced cellularity (Fig. 1b, right). HSC numbers in the liver and spleen were found normal compared to controls, and bone marrow HSCs recover to normal levels in 70 days-old *Rank*[cre/+];*Csf1r*[fl/−] mice (Supplementary Fig. 1b).

Molecularly, HSCs from 3-week-old *Rank*[cre/+];*Csf1r*[fl/−] mice show 515 genes positively and 46 genes negatively, significantly deregulated compared to controls (Supplementary Fig. 1c, Supplementary Data 1). Functional annotation clustering analysis included enrichments in bone mineralization, cell-cell adhesion, and collagen metabolic process suggesting differing HSC niche communication in *Rank*[cre/+];*Csf1r*[fl/−] mice compared to controls (Fig. 1c, Supplementary Data 2).

The frequency of phenotypic HSPCs in the blood and spleen is increased (Fig. 1d, e), which is accompanied by elevated hematopoietic colony formation potential in the spleen of *Rank*[cre/+];*Csf1r*[fl/−] mice (Fig. 1e). However, non-competitive transplantation revealed normal repopulation activity (kinetics and extent) of *Rank*[cre/+];*Csf1r*[fl/−] bone marrow stem and progenitor cells (HSPCs). HSCs (KSL Slam) and HSPCs (KSL) are equally reduced in the bone marrow of *Rank*[cre/+];*Csf1r*[fl/−] mice (Fig. 1b) so that HSPCs from *Rank*[cre/+];*Csf1r*[fl/−] and control mice contain the same number of HSCs, and hematopoietic function is comparable. This suggests that cellularity but not per cell potential is decreased in *Rank*[cre/+];*Csf1r*[fl/−] mice (Fig. 1f). Cell cycle analysis revealed increased cycling activity of HSCs in the bone marrow of *Rank*[cre/+];*Csf1r*[fl/−] mice but not in the spleen in situ, suggesting a compensatory mechanism to adjust to normal numbers of HSCs in *Rank*[cre/+];*Csf1r*[fl/−] mice (Supplementary Fig. 1d). We conclude that HSC numbers are reduced in the bone marrow of *Rank*[cre/+];*Csf1r*[fl/−] mice but exhibit normal functionality.

### Unperturbed generation of definitive HSCs

Reduced HSC numbers may be a consequence of their impaired generation. However, normal frequencies and numbers of pre-HSCs[25–29] are present in the aorta-gonad-mesonephros (AGM) region of *Rank*[cre/+];*Csf1r*[fl/−] embryos at E10.5 during gestation (Fig. 2a). Further, wildtype-like numbers of HSCs are present in the liver and spleen at E17.5 (KSL CD48[−] CD150[+] [Slam][9], (Fig. 2b, Supplementary Fig. 2a). In contrast, HSC numbers are reduced in the bone marrow of *Rank*[cre/+];*Csf1r*[fl/−] mice (Fig. 2b), suggesting that the generation of HSCs is not altered in the absence of embryonic myeloid cells in situ, but, instead the setup of the stem cell pool in the bone marrow is impaired. Strikingly, hematopoietic progenitor cell migratory capacities seem normal since KSL frequencies and numbers in the bone marrow of *Rank*[cre/+];*Csf1r*[fl/−] mice are comparable to wildtype controls (Supplementary Fig. 2b).

### Defective establishment of a definitive HSC pool in the absence of embryonic myeloid cells

To further test migration properties of HSCs, their numbers in bone marrow, liver and spleen were analyzed in newborn mice (Fig. 2c). HSPC numbers are reduced in the bone marrow and spleen but increased in the liver of newborn mice. Alterations in numbers are consistent with changes in the frequency of growth factor responsive cells in all three organs (Fig. 2d), suggesting that HSPC migration is altered through embryonic myeloid cells that are missing in *Rank*[cre/+];*Csf1r*[fl/−] mice.

CXCR4 is important for the migration and retention of HSCs in the bone marrow[30–32] and, consistent with reduced HSC cellularity, the abundance of CXCR4 expression on HSCs is decreased in the bone marrow of newborn mice of *Rank*[cre/+];*Csf1r*[fl/−] mice compared to controls (Fig. 2e). To test for corresponding overall reduced reconstitution activity as proxy for decreased stem cell numbers, whole bone marrow cells from newborn *Rank*[cre/+];*Csf1r*[fl/−] or control mice were transplanted into non-myeloablated congenic recipient mice (Fig. 2f). Repopulation of blood neutrophils was blunted in terms of the kinetic and overall extent of donor-cell contribution and a reduced contribution of *Rank*[cre/+];*Csf1r*[fl/−] cells to the HSC pool was measured 18 weeks after transplantation, confirming a reduced frequency of functional HSCs in the bone marrow of newborn *Rank*[cre/+];*Csf1r*[fl/−] mice. HSPCs from *Rank*[cre/+];*Csf1r*[fl/−] mice further displayed reduced migratory capacities towards CXCL12 (SDF-1) in vitro (Fig. 2g), and whole bone marrow cells

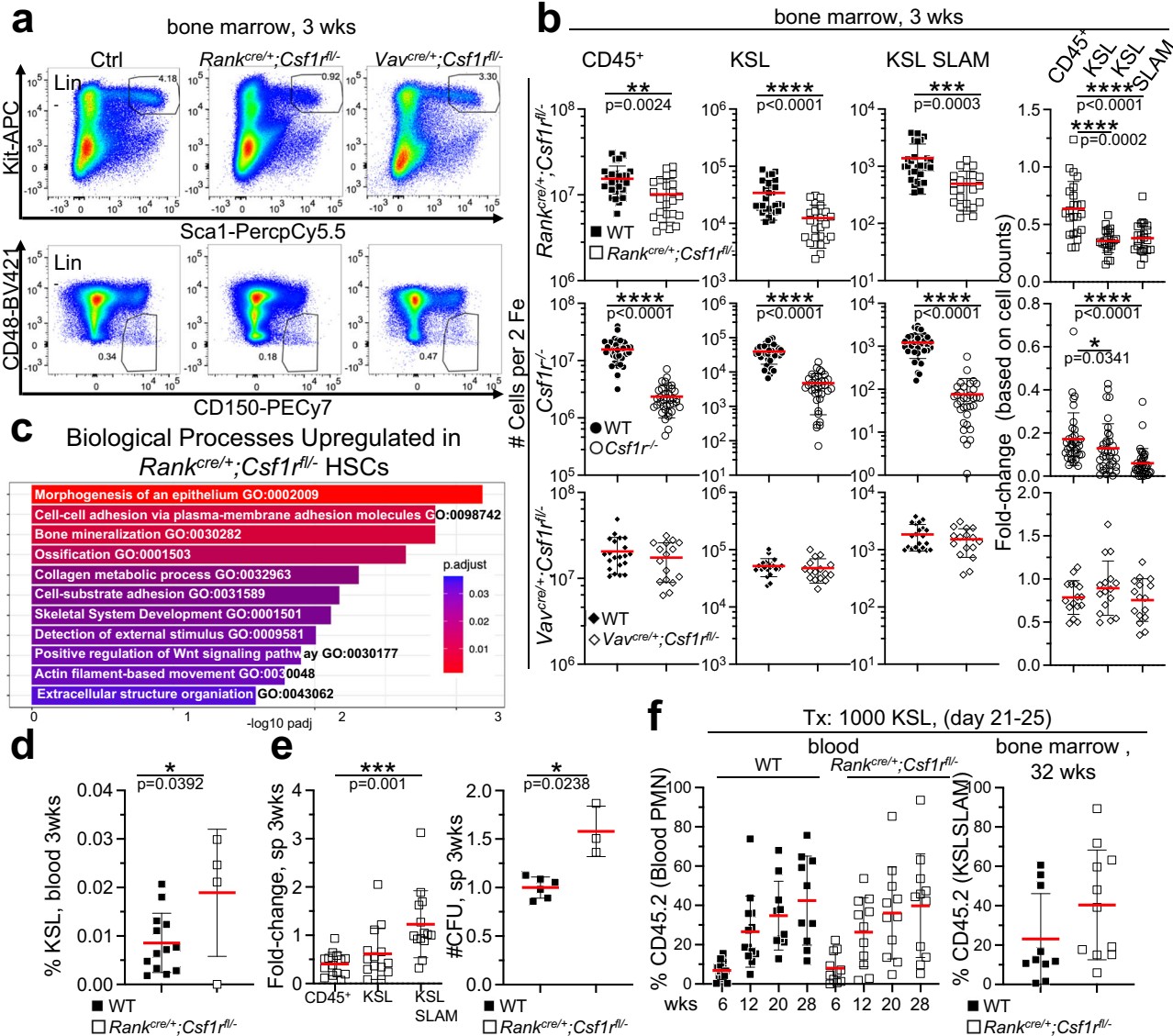

**Fig. 1 | Lack of embryonic myeloid cells results in decreased numbers of functional HSCs. a** Dot plots depict bone marrow cells of indicated mice at 3 weeks (18–32-day-old) of age resolved for the expression of indicated antigens. dapi-negative singlets were gated on lineage negative cells (Lin = CD3 CD19 NK1.1 Ter119 CD11b Gr1 B220) followed by Kit+ Sca1+ (KSL) gating. KSL cells were further sub-divided into CD48- CD150+ KSL SLAM cells. **b** Quantification of leukocytes (CD45), HSPCs (KSL), and HSCs (KSL Slam) as gated in a from mice of indicated genotypes in the bone marrow of 3-week-old mice (left). A two-sided unpaired Student's t-test was used. (*n* = 51, 11 biological replicates for *Rank^cre/+;Csf1r^fl/-*, *n* = 72, 16 biological replicates for *Csf1r^-/-*, *n* = 37, 6 biological replicates for *Vav^cre/+;Csf1r^fl/-*). Fold-change comparisons between indicated populations in *Rank^cre/+;Csf1r^fl/-* mice compared to controls (right). Fold-changes were calculated by dividing the indicated cell types from *Rank^cre/+;Csf1r^fl/-* mice to the experimental average of the wildtype numbers For fold-change comparisons (right), a Mann–Whitney U test was performed to assess statistical significance between individual groups with an expected normal distribution. **c** Plot shows biological processes that are enriched in genes up-regulated in HSCs (KSL Slam) from *Rank^cre/+;Csf1r^fl/-* mice compared to wildtype controls (19–21-day-old). Over-representation analysis with a p-adjusted cutoff of 0.05 and subsequent semantic clustering was performed for 515 up-regulated genes using the

gene ontology for biological processes from MSigDB. Colors indicate the p-adjusted value. A one-sided hypergeometric test was performed, with significance adjusted for multiple testing using the Benjamini-Hochberg method. **d** Plot depicts frequencies of KSL in the blood of 3-week-old (18–23-day-old-mice) *Rank^cre/+;Csf1r^fl/-* mice and controls. A two-sided unpaired Student's t-test was conducted for statistical analysis. (*n* = 17, 3 biological replicates) **e** Fold-change of leukocytes (CD45), HSPCs (KSL), and HSCs (KSL Slam) from the spleen of 3-week-old *Rank^cre/+;Csf1r^fl/-* mice compared to wildtype mice is shown (left). A Mann–Whitney U test was performed for statistical analysis. (*n* = 14, 6 biological replicates) Colony formation from splenocyte suspension from indicated genotypes are depicted (right). A Mann–Whitney U test was performed. (*n* = 12, 2 biological replicates) **f** Plot shows donor cell contribution by HSPCs (KSL) from *Rank^cre/+;Csf1r^fl/-* mice and control donors (21–25-day-old) to blood neutrophils (PMN) at indicated time points after transplantation into non-conditioned recipient mice (*Rag2^-/-;Il2rg^-/-;Kit^W41/W41*, RgW41, modified from[34], left). A two-sided unpaired Student's t-test was conducted for statistical analysis. (*n* = 24, 2 biological replicates) Donor cell contribution to HSCs (right). Each dot represents the value for one mouse in b, d-f. Source data are provided as a Source Data file.

from newborn *Rank^cre/+;Csf1r^fl/-* mice harbored a significantly decreased ability to home into the bone marrow of newborn wildtype recipient mice 16 h after intrahepatic transfer (Fig. 2h). We conclude that HSC retention in the bone marrow is impaired in the absence of embryonic myeloid cells.

## A population of functional embryonic macrophages resides in the bone marrow

A lineage-tracing approach was chosen and *Rank^cre/+;R26.eYFP^fl/+* mice were used to identify embryonic macrophages possibly contributing to the phenotype. Mice carrying *Rank-cre* in combination with the

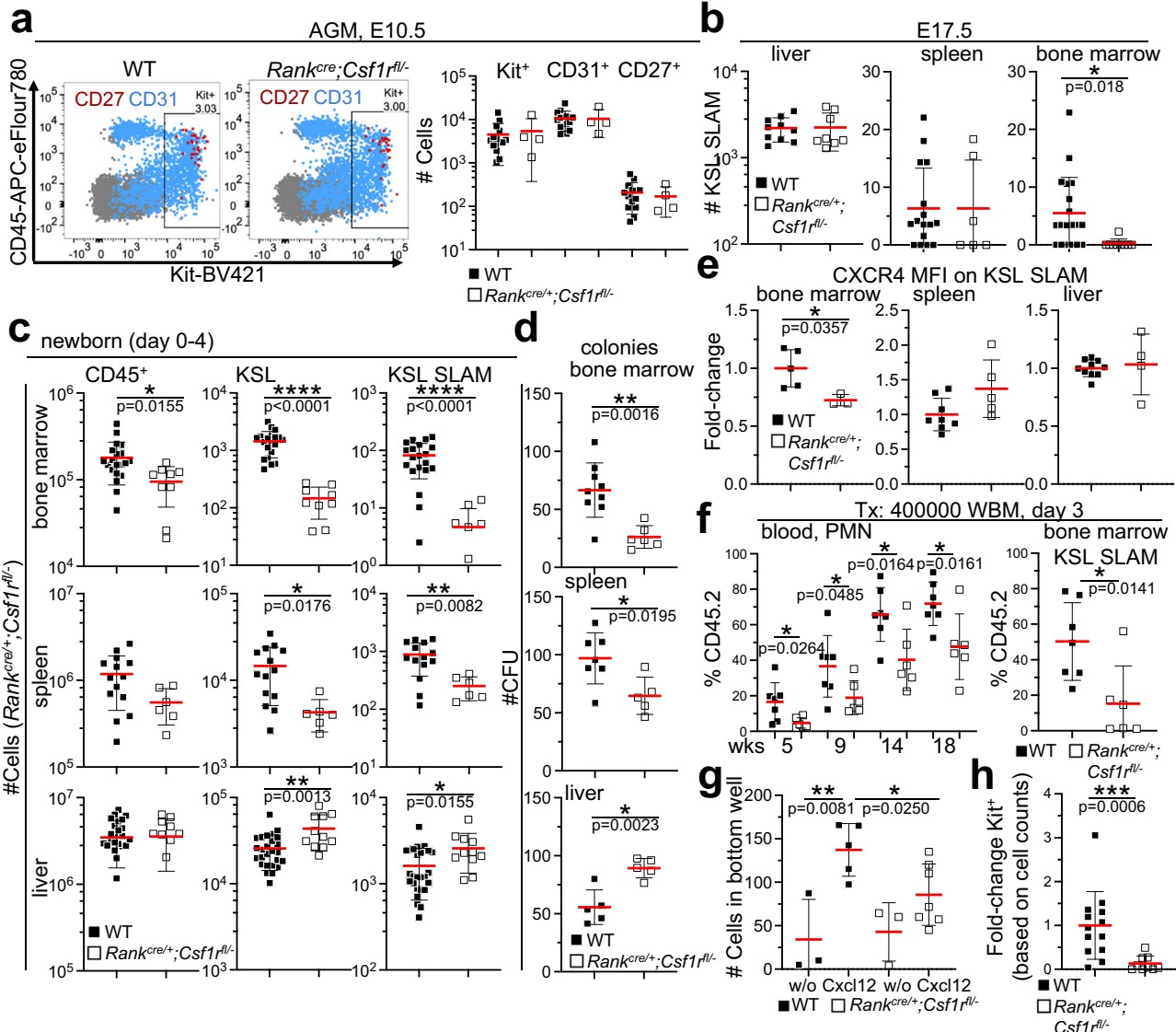

**Fig. 2 | Normal emergence of definitive HSCs but defective translocation to the bone marrow. a** Dot plots resolve cells of the aorta-gonad mesonephros (AGM) region of E10.5 embryos of indicated genotypes for the expression of CD45 and Kit. Cells expressing CD31 (blue) or CD27 (red) are superimposed into the dot plots (left). Quantification of Kit⁺, CD31⁺, and CD27⁺ cells per AGM (right). A two-sided unpaired Student's t-test was conducted for statistical analysis. (*n* = 18, 2 biological replicates). **b** Plots show numbers of HSCs (KSL Slam) in the liver, spleen, and bone marrow (2 femura) of E17.5 embryos of indicated genotypes. A two-sided unpaired Student's t-test was conducted for statistical analysis. (*n* = 18, 3 biological replicates for fetal liver, *n* = 21, 3 biological replicates for fetal spleen, n = 27, 4 biological replicates for fetal bone marrow). **c** Plots show the number of leukocytes (CD45), HSPCs (KSL), and HSCs (KSL Slam) in the bone marrow, spleen, and liver of newborn mice of indicated genotypes. A two-sided unpaired Student's t-test was used. (*n* = 29, 4 biological replicates for bone marrow, *n* = 21, 4 biological replicates for spleen, *n* = 37, 6 biological replicates for liver). **d** Graphs show in vitro colony formation from bone marrow, spleen, and liver cell suspensions of newborn mice (0–4-day-old) of indicated genotypes. A two-sided unpaired Student's t-test was used. (*n* = 15, 3 biological replicates for bone marrow, *n* = 12, 3 biological replicates for spleen, *n* = 14, 3 biological replicates for liver). **e** Graphs show the fold-change of

CXCR4 mean fluorescence intensity (MFI) on HSCs in the bone marrow, spleen and liver of newborn (0–4-day-old) mice of indicated genotypes. Fold-change was calculated by dividing the individual CXCR4 MFI value of *Rank^{cre/+};Csf1r^{fl/-}* HSCs to the experimental average of wild-type CXCR4 MFI values per organ. A Mann–Whitney U test was performed. (*n* = 8, 2 biological replicates for bone marrow, *n* = 13, 3 biological replicates for spleen, n = 14, 3 biological replicates for liver) **f** Graph shows donor cell contribution to blood neutrophils (PMN) at indicated time points after transplantation of unfractionated bone marrow cells from 2-days-old donor mice of indicated genotypes (left). Graph shows the contribution of donor-derived cells to HSCs (KSL Slam) in the bone marrow at the time point of analysis (right). A two-sided unpaired Student's t-test was used. (*n* = 13, 2 biological replicates) **g** Plot depicts the number of CXCL12-mediated migration of liver-derived HSPCs (KSL) from 1–4-day-old mice, indicated genotypes after 2.5 h. A two-sided unpaired Student's t-test was used. (*n* = 9, 3 biological replicates) **h** Fold-change of 1-day-old donor-derived Kit⁺ cells that have migrated to the bone marrow 16 h after transfer of whole bone marrow cells from newborn (0–1-day-old) mice of indicated genotypes into the liver of newborn wild-type recipient mice. A Mann–Whitney U test was performed. (*n* = 22, 2 biological replicates) Each dot represents the value for one mouse throughout the figure. Source data are provided as a Source Data file.

*Rosa26.eYFP^{fl/+}* reporter alleles allow for lineage tracing of embryonic macrophages because there is no expression of the CRE-recombinase in adult hematopoietic stem or progenitor cells[19,20]. Reporter-positive cells contribute to CD11b^{lo} F4/80^{hi} macrophages but not to CD11b⁺ F4/80^{lo} monocytes or other myeloid cell types in the bone marrow (Fig. 3a,

Supplementary Fig. 3a), identifying a population of hitherto unknown embryo-derived macrophages in the murine bone marrow.

Lineage-tracing over lifetime reveals that embryonic macrophages constitute a large proportion of bone marrow F4/80^{hi} macrophages at birth but decline during aging (Fig. 3b, c). Cellular longevity

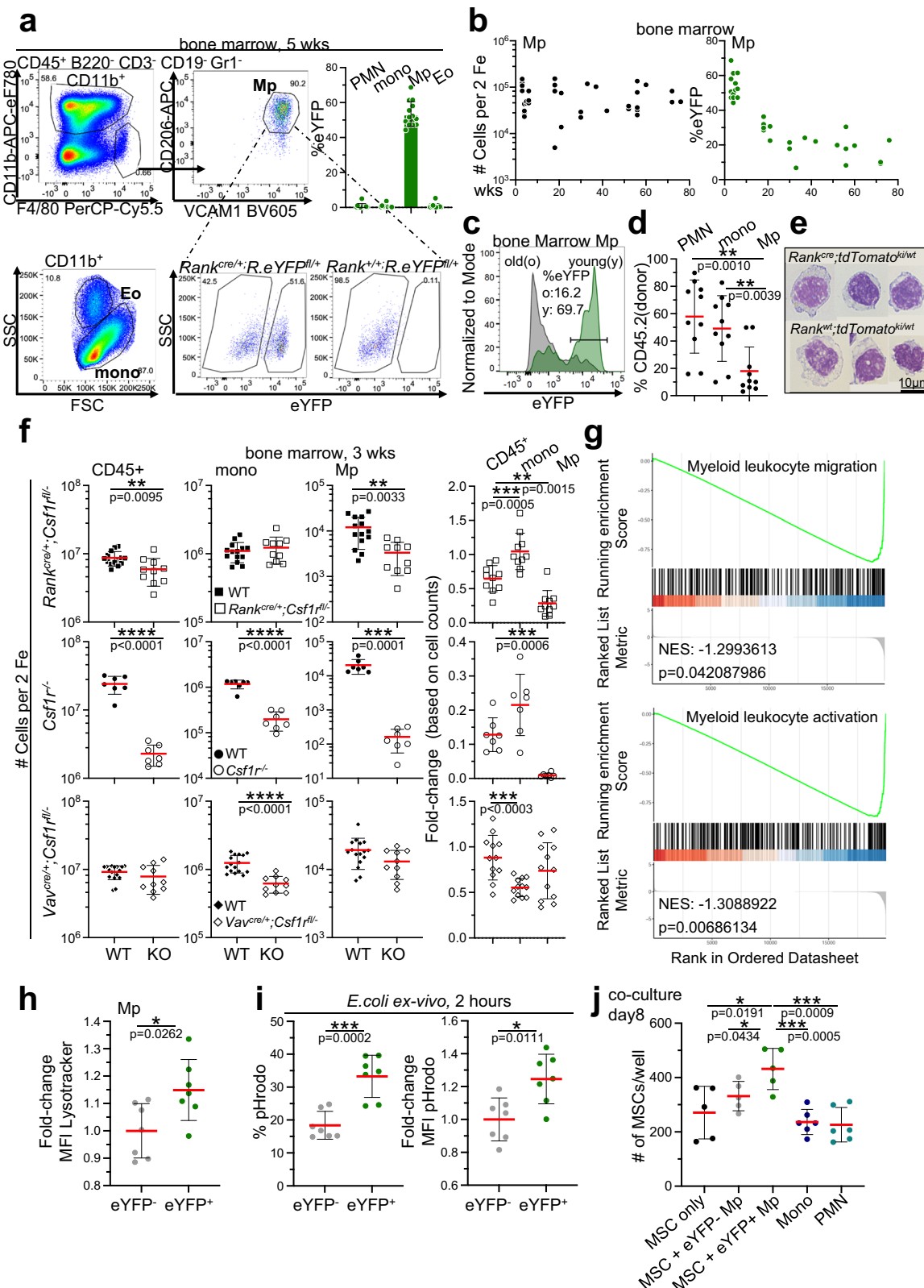

is a hallmark of embryo-derived macrophages that reside in many tissues[19,33]. To address the turn-over of bone marrow embryonic macrophages, HSPCs from wild-type mice were transplanted into non-myeloablated recipient mice that were chosen to avoid cellular damage through the conditioning regimen (Fig. 3d)[34]. This allows to detect the contribution of donor-derived cells under steady-state conditions. A slow turnover of bone marrow embryonic macrophages

was confirmed because neutrophils and monocytes, were repopulated by donor-derived cells to a higher extent compared to F4/80+ macrophages at the time point of analysis (Fig. 3d). Donor cells contribute to HSC-derived neutrophils and monocytes to equal levels 32 weeks after transplantation, whereas the contribution of donor cells to F4/80hi macrophages that include a large proportion of embryo-derived cells is significantly reduced, evidencing their slow turn-over.

**Fig. 3 | A population of embryonic macrophages in the bone marrow. a** Dot plots show gating for myeloid cells in the bone marrow of *Rank^cre/+;R26.eYFP^fl/+* mice. For macrophages, dapi-negative singlets were gated on CD45^+ cells. B220, CD3, and CD19 postitive cells were further subdivided by Ly6C and Gr1. Gr1^low cells were further gated on CD11b vs F4/80. Monocytes and Eosinophils were identified as CD11b^+ cells separated by scatter characteristics as shown. Macrophages were gated on F4/80^+ cells and further subdived by CD206^+ and VCAM1^+. Double-positive population were identified as macrophages and analyzed for eYFP expression. Bar graph shows the quantification of eYFP frequencies within indicated myeloid cells in the bone marrow. Mp macrophages, Eo eosinophils, mono monocytes, PMN polymorph nuclear neutrophils. **b** Quantification of F4/80^hi CD206^+ VCAM1^+ cells per 2 femura (left). Contribution of eYFP^+ cells to F4/80^hi CD206^+ VCAM1^+ cells (right). **c** Histograms show the frequency of eYFP expression in 3-week-old (young, green) and 83-week-old (old, grey) macrophages of *Rank^cre/+;R26.eYFP^fl/+* bone marrow. Mp=macrophages. **d** Repopulation of indicated myeloid cell populations with donor-derived cells in the bone marrow of mice that have received congenic bone marrow cells 32 weeks before. A two-sided unpaired Student's t-test was used. (*n* = 10, 3 biological replicates). **e** Giemsa staining of tdTomato^+ and tdTomato^- bone marrow macrophages cytospins, from 6–9-week-old mice, isolated as depicted in **a**. (*n* = 3, 2 biological replicates). **f** Plots show the number of leukocytes (CD45), monocytes (mono), and macrophages (Mp) in the bone marrow of 3-week-old mice of indicated genotypes (left). A two-sided unpaired Student's t-test was used. (*n* = 24, 5 biological replicates for *Rank^cre/+;Csf1r^fl/-*, *n* = 14, 4 biological replicates for *Csf1r^-/-*, *n* = 30, 5 biological replicates for *Vav^cre/+;Csf1r^fl/-*). Fold-change

comparisons between indicated bone marrow cell populations of wild type and *Rank^cre/+;Csf1r^fl/-* or *Csf1r^-/-* or *Vav^cre/+;Csf1r^fl/-* mice (right). A Mann–Whitney U test was performed for statistical analysis. (*n* = 14, 6 biological replicates) **g** Gene set enrichment analysis (GSEA) of genes differentially expressed by eYFP^+ embryonic and eYFP^- adult bone marrow macrophages using gene sets from MSigDB v2023.2Mm database. Normalized enrichment score (NES) and by Benjamini-Hochberg method adjusted p-values are shown for GO terms myeloid leukocyte migration and myeloid leukocyte activation. **h** Fold-change of mean fluorescence intensity (MFI) of Lysotracker-positive cells within embryonic (eYFP^+) and adult-derived (eYFP^-) bone marrow macrophages from 4–7-week-old mice. Fold-change was calculated by dividing individual MFI values of eYFP^+ macrophages to the experimental average of the eYFP^- MFI values in each experiment. A Mann–Whitney U test was performed for statistical analysis. (*n* = 7, 3 biological replicates) **i** Frequency of pHrodo positive cells within embryonic (eYFP^+) and adult-derived (eYFP^-) bone marrow macrophages after a 2 h ex vivo incubation with *E.coli.* bio-particles (left). A two-sided unpaired Student's t-test was used. (n = 7, 3 biological replicates). Fold-change of pHrodo mean fluorescence intensity (MFI) values of eYFP^+ and eYFP^- macrophages (right) Fold-change was calculated by dividing individual MFI values of eYFP^+ macrophages to the experimental average of the eYFP^- MFI values in each experiment. A Mann–Whitney U test was performed for statistical analysis. (*n* = 7, 3 biological replicates). 4–7-week-old mice were used. **j** Quantification of MSC cell numbers after culturing with or without bone marrow myeloid cells for 8 days. A two-sided unpaired Student's t-test was used. (*n* = 5, 3 biological replicates). Source data are provided as a Source Data file.

Morphologically, embryonic and adult-derived F4/80^hi macrophages are identical (Fig. 3e). Consistent with growth factor receptor-dependency of mixed F4/80^hi macrophage populations in other tissues[20] and embryo-derived osteoclasts in the bone marrow[19], the total pool of F4/80 positive cells is significantly reduced in *Rank^cre/+;Csf1r^fl/-* mice at 3 weeks of age, suggesting that embryonic but not adult-derived F4/80^hi macrophages or monocytes in the bone marrow depend on CSF1R signaling for their generation (Fig. 3f). Bone marrow macrophage numbers normalize in adult mice, most likely based on reconstitution of this cellular compartment by definitive HSC-derived cells (Supplementary Fig. 3b).

To determine potential differences in their cellular identity, transcriptomes of embryonic and adult-derived F4/80^hi macrophages and monocytes were generated. Principal component analysis (PCA) revealed a high degree of similarity between embryonic and adult macrophages (eYFP^+ vs eYFP^-) along the first principal components, while Ly6C^lo and Ly6C^hi monocytes clustered substantially further apart (Supplementary Fig. 3c). Upon focusing on macrophages, however, the PCA recovers differences between eYFP^+ and eYFP^- cells, thus this analysis suggests transcriptional differences. Gene expression analysis revealed 432 significantly differentially regulated genes between eYFP^+ and eYFP^- macrophages with the majority of genes being down-regulated in embryonic macrophages (eYFP^+) (Supplementary Fig. 3d). Gene set enrichment analysis (GSEA) revealed significant enrichment of these differentially expressed genes in gene ontologies like myeloid leukocyte migration (GO:0097529) and myeloid leukocyte activation (GO:0002274) (Fig. 3g, Supplementary Fig. 3e). Genes involved in myeloid leukocyte migration, including *Sell*[35] and *Ccr2*[36], are down-regulated in embryonic macrophages, providing a possible explanation for their settledness. Furthermore, down-regulation of *Lbp*, an enhancer of TLR4 responses upon LPS trigger[37], and *Cst7* that positively regulates phagocytosis[38], suggests that the origin of F4/80+ macrophages has an impact on their effector function. Consistently, over-representation analysis revealed an enrichment in gene sets associated with cell migration and activation in eYFP^+ embryonic macrophages (downregulated genes, Supplementary Fig. 3f, Supplementary Data 3). Results from up-regulated genes in eYFP^+ macrophages, in contrast, are associated with endocytosis, actin filament organization and inflammatory response, suggesting a role for embryonic macrophages in HSC niche generation and regulation.

Together, despite the identical cell surface phenotype of eYFP^+ embryonic and eYFP^- adult bone marrow macrophages, the transcriptional differences suggest a significant influence of ontogeny on macrophage functions, including the development of bone marrow tissue.

## The developmental origin of macrophages impacts their functionality

Functional differences were indicated by the distinct transcriptional profiles of bone marrow macrophages of embryonic and adult origin. To test for such functional differences, lysosome organelle content and phagocytosis activity of embryonic (eYFP^+) and adult-derived (eYFP^-) F4/80^hi macrophages was analyzed using *Rank^cre/+;R26.eYFP^fl/+* mice. Embryonic F4/80^hi macrophages display a higher content of acidic organelles compared to adult-derived macrophages, suggesting an increased digestive activity (Fig. 3h). Consistently, phagocytic activity is increased, both in terms of the frequency of cells that take up bioparticles, but also in terms of the quantity of bioparticles taken up per macrophage (Fig. 3i). Together, this suggests that the functionality of bone marrow-resident macrophages depends on their developmental origin. To test for a supportive role in niche cell differentiation, MSCs were co-cultured with embryonic- and adult-derived macrophages, monocytes, or neutrophils, and only macrophages of embryonic origin proved supportive for the maintenance of MSCs (Fig. 3j), suggesting an indirect effect of macrophages on HSC numbers via the niche.

## Embryonic myeloid cells orchestrate the development of a normal stem cell niche in the bone marrow

An organized bone marrow niche is key for its seeding with definitive HSCs and we have assessed the non-hematopoietic niche cell populations[39] in *Rank^cre/+;Csf1r^fl/-* newborns and control mice. Mesenchymal stromal cells are significantly decreased in the bone marrow of *Rank^cre/+;Csf1r^fl/-* newborn mice but recover to wild type levels by 3 weeks of age (Fig. 4a, b, Supplementary Fig. 4a), suggesting a rapidly adjusting microenvironment during neonate HSC pool establishment. Histological analysis underscores the severe decrease of CD51 positive mesenchymal cells (Fig. 4c). Endothelial cells and osteoprogenitors show no decrease in frequencies compared to controls (Fig. 4a–c). Consistently, endothelial cell growth and tube formation is unperturbed in neonate *Rank^cre/+;Csf1r^fl/-* mice (Fig. 4d),

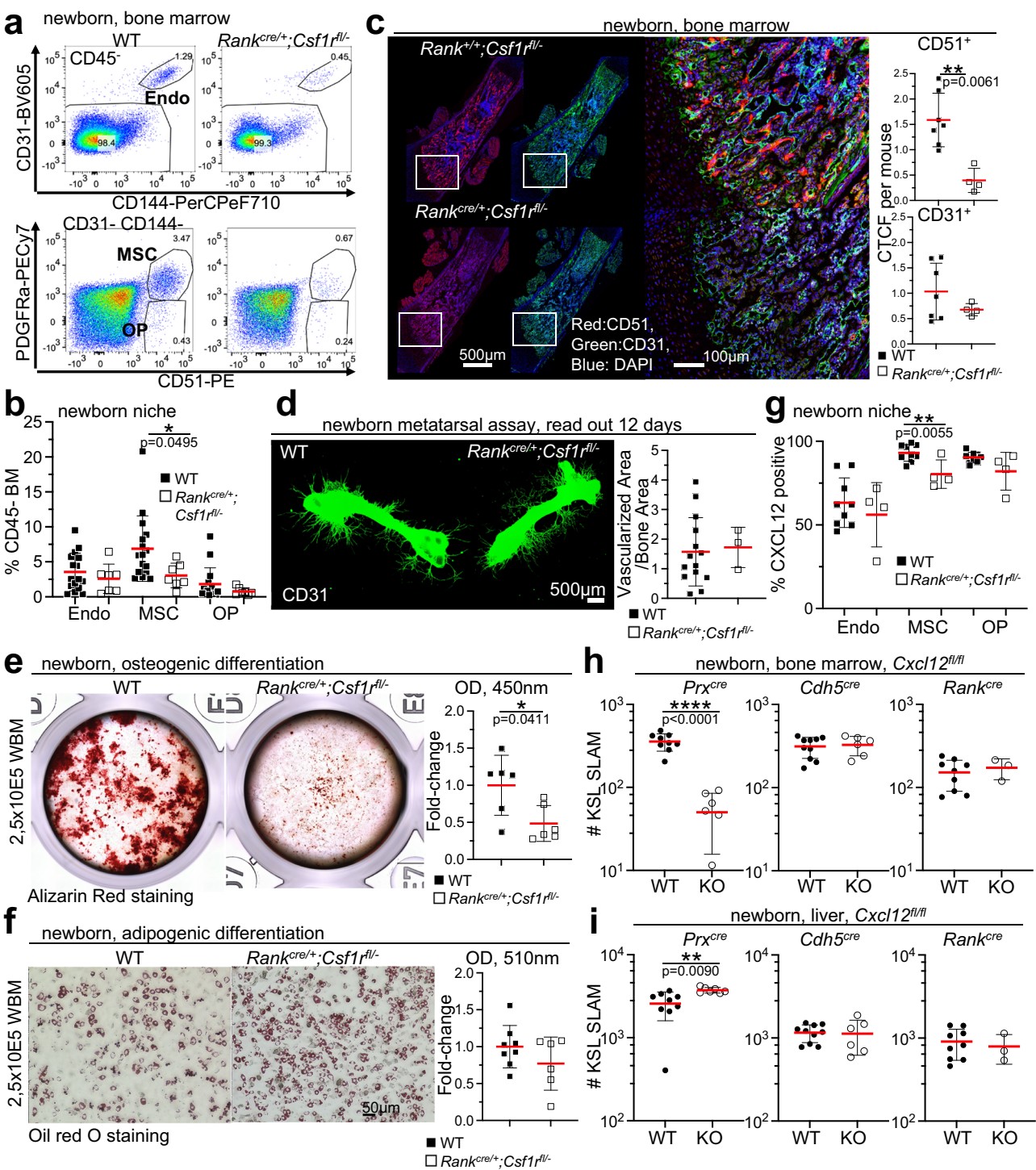

evidencing normal development and function of endothelial cell populations. In contrast, osteogenic potential of bone marrow cells from *Rank^{cre/+};Csf1r^{fl/-}* mice is severely blunted (Fig. 4e), pinpointing at a potential role for embryonic macrophages in establishing osteogenic propensity in MSCs. Consistently, co-culture of MSCs from newborn mice with embryonic but not with adult macrophages enhances osteogenic potential compared to MSC cultures alone (Supplementary Fig. 4b). However, the defective differentiation potential of mesenchymal progenitors is restricted to the osteogenic fate since adipogenic differentiation is unperturbed (Fig. 4f). We conclude that MSC numbers and their differentiation potential depend on embryonic myeloid cells.

To determine specific molecular interactions potentially mediating differential effects of embryonic versus adult macrophages on MSC biology, we investigated receptor-ligand pair RNA-seq derived expression ([39], https://github.com/hoefer-lab/CellInteractionScores) between MSCs and macrophages of different developmental origin. In silico predicted molecular interactions are associated with remodeling of extracellular matrix and cell signaling (Supplementary Fig. 4c, Supplementary Data 4), suggesting an influence of macrophages on HSC niche formation via extracellular matrix organization. Specifically, *Lrg1* is expressed by adult but not embryonic macrophages and it regulates the stiffness of extracellular matrix components[40], and the expression of its receptor, *Tgfbr2* on MSCs is a key regulator for the

**Fig. 4 | Establishment of the stem cell niche in the absence of embryonic myeloid cells. a** Dot plots display non-hematopoietic niche cells in the bone marrow of newborn (2 days) $Rank^{cre/+};Csf1r^{fl/-}$ and control mice. From dapi-negative singlets leukocytes (CD45-) and red blood cells (Ter119) were excluded and endothelial cells (endo) identified by cell surface expression of CD31 and CD144. CD31 negative cells were further subdevided by CD51 and PDGFRa to identify mesenchymal stromal cells, MSC and osteoprogenitors, OP. **b** Frequencies of niche cells within CD45- bone marrow cells identified as shown in a. A two-sided unpaired Student's t-test was used. ($n = 23$, 5 biological replicates). **c** Representative photographs of histological sections of newborn (1-day-old) $Rank^{cre/+};Csf1r^{fl/-}$ and control mice stained with antibodies specific for MSCs (CD51, red), endothelial cells (CD31, green), and DNA (DAPI, blue) (left). Quantification of corrected total cell fluorescence (CTCF) of CD51 (right, top) and CD31 (right, bottom) positive cells. Each dot represents the value for one mouse. A Mann–Whitney U test was performed for statistical analysis. ($n = 11$ [7 wt, 4 mutant], 3 regions of interest were imaged in each bone, 3 independent experiments). **d** Photographs show CD31-stained cultivated metatarsals from newborn (3-day-old) $Rank^{cre/+};Csf1r^{fl/-}$ or control mice (left, 12 days). Plot shows the ratio vascularized area / bone area (right). A two-sided unpaired Student's t-test was used. ($n = 17$, 3 biological replicates). **e** Photographs show alizarin red stained osteogenic cultures of

unfractionated bone marrow cells from newborn (4-day-old) $Rank^{cre/+};Csf1r^{fl/-}$ and control mice (left, 11 days). Graph shows the quantification of osteogenic spots (alizarin red absorbance at 450 nm). A Mann–Whitney U test was performed for statistical analysis. ($n = 12$, 2 biological replicates). WBM whole bone marrow. **f** Photographs show Oil red O stained adipogenic cultures of unfractionated bone marrow cells from newborn (3-day-old) $Rank^{cre/+};Csf1r^{fl/-}$ and control mice (left, 7 days). Graph shows the quantification of Oil red O absorbance (510 nm). A Mann–Whitney U test was performed for statistical analysis. ($n = 14$, 2 biological replicates). **g** Frequencies of CXCL12 positive niche cells (intracellular staining) in bone marrow of newborn (2–4-day-old) $Rank^{cre/+};Csf1r^{fl/-}$ and controls. A two-sided unpaired Student's t-test was used. ($n = 13$, 3 biological replicates). **h** Plots show numbers of KSL Slam HSCs in the newborn (0–4-day-old) bone marrow of mice of indicated genotypes. A two-sided unpaired Student's t-test was used. ($n = 16$, 3 biological replicates for $Prx^{cre/+};Cxcl12^{fl/fl}$, n = 16, 3 biological replicates for $Cdh5^{cre/+};Cxcl12^{fl/fl}$, $n = 13$, 2 biological replicates for $Vav^{cre/+};Csf1r^{fl/-}$). **i** Plots show numbers of KSL Slam HSCs in the newborn (0–4-day-old) liver of mice of indicated genotypes. A two-sided unpaired Student's t-test was used. ($n = 16$, 3 biological replicates for $Prx^{cre/+};Cxcl12^{fl/fl}$, n = 16, 3 biological replicates for $Cdh5^{cre/+};Cxcl12^{fl/fl}$, $n = 13$, 2 biological replicates for $Vav^{cre/+};Csf1r^{fl/-}$). Each dot represents the value for one mouse in figure parts c–i. Source data are provided as a Source Data file.

normal development of long bones[41]. Also *Mfge8* is specifically expressed by adult macrophages and supports tissue repair[42]. Its molecular interaction partners, *Itgb3* and *Pdgfrb*, are also linked to Mfge8-mediated regulation of local innate immune responses[43] and to the differentiation potential of MSCs[44], respectively, suggesting that macrophages of distinct origin shape the stem cell niche for the specific needs at different time points in life.

Microenvironment-derived CXCL12 is critical during development to attract definitive HSCs to the bone marrow niche and it is expressed by nestin receptor positive[11] or leptin receptor positive cells[13]. CXCL12 is expressed by osteoblast precursors and, to higher levels, by MSCs that significantly upregulate the expression by 3 weeks of age (Supplementary Fig. 4d). In the absence of embryonic myeloid cells, CXCL12 expression is decreased by neonate MSCs but not endothelial cells or osteoblast precursors (Fig. 4g). To test whether CXCL12 expressed by MSCs, but not by endothelial cells or embryonic myeloid cells, affects HSC numbers, $Prx^{cre/+};Cxcl12^{fl/fl}$, $Cdh5^{cre/+};Cxcl12^{fl/fl}$, and $Rank^{cre/+};Cxcl^{fl/fl}$ mice were generated. $Prx^{cre/+}$ was chosen because $Lepr^{cre/+}$-mediated recombination of a floxed allele is lacking in neonate niche cells (Supplementary Fig. 4e). Definitive HSC numbers in newborn mice are independent of CXCL12-production by embryonic myeloid and endothelial cells (Fig. 4h, I and Supplementary Fig. 4f). However, the phenotype of $Rank^{cre/+};Csf1r^{fl/-}$ mice is completely recapitulated in $Prx^{cre/+};Cxcl12^{fl/fl}$ mice, suggesting an important role for embryonic myeloid cells in the development of a functional stem cell niche based on MSC numbers and function in vivo.

## Discussion

We show here that embryonic myeloid cells are crucial for the establishment of a normal bone marrow niche amenable to attracting and hosting definitive HSCs during development. In the absence of embryonic myeloid cells in the bone marrow, mesenchymal stromal cells are severely reduced and show a blunted production of CXCL12, a chemokine critical for alluring definitive HSCs. As a consequence, the numbers of definitive HSCs are significantly reduced in neonate and 3-week old mice in the absence of embryonic myeloid cells; however, on a per-cell basis stem cells remain fully functional. We thus identify a cellular regulator orchestrating the composition of the bone marrow hematopoietic stem cell niche that is key to establish a normal-sized pool of stem cells important for lifelong blood cell supply.

In adult mice bone marrow macrophage-lineage cells regulate HSC retention, maintenance, and response to inflammatory challenges[45,46]. This regulation has been suggested to be mediated indirectly via defined niche cells including mesenchymal stromal cells, endothelial cells, and osteoblast progenitors. There is strong evidence that also during development pro-inflammatory signaling instigates

HSC emergence in zebrafish[47,48], and macrophages play a direct role in providing quality assurance for the emerging HSC pool[49]. In the mouse, elegant explant culture systems have revealed that phagocytic mononuclear cells are important for the generation of normal numbers of transplantable stem cells, further suggesting a niche-controlled mechanism in the emergence of HSCs[50]. However, in mice that lack embryonic but not adult bone marrow myeloid cells[19,20], we find normal numbers of pre-HSCs at E10.5 in the aorta gonad mesonephros (AGM) region and HSCs at E17.5 in the liver, suggesting normal emergence of HSCs and trafficking from the AGM to the fetal liver. In contrast, at the same time-point, HSC numbers are reduced in the bone marrow, pointing towards a blunted relocation of HSCs into the bone marrow that relies on embryonic myeloid cell-mediated effects. These results further suggest a functional dichotomy between embryonic- and adult-derived myeloid cells in supporting HSC emergence versus relocation.

Tissue-resident macrophages (TR-Mp) are long-lived[33], and originate to large parts from embryonic progenitors[19,20]. Cues from the microenvironment impact the transcriptional profile that determines cellular identity and function of tissue-resident macrophages[51,52]. Within the pool of bone marrow mononuclear phagocytes only osteoclasts have been assigned to embryonic origin based on lineage-tracing and functional experiments[19]. However, using lineage-tracing mouse tools we show here that also F4/80 positive tissue macrophages but not other myeloid cells in the bone marrow are of embryonic origin. Embryonic versus adult macrophages display differences in gene expression, suggesting distinct functional roles within this tissue. Consistently, embryonic macrophages harbor increased acidic organelles, and exhibit enhanced phagocytic activity compared to macrophages of adult origin, providing evidence that a distinct ontogenetic origin impacts the function of bone marrow macrophages.

Embryo-derived myeloid cells are identified as mediators for the establishment of a normal bone marrow microenvironment that supports the seeding of HSCs in newborn mice. Lack of embryonic myeloid cells in the bone marrow leads to decreased frequencies of MSCs, hinting at a compromised niche formation for HSC homing and retention. Other non-hematopoietic niche cells, such as endothelial cells and adipogenic precursors were unaltered in frequencies. Further, quantitative functional assays revealed no differences of endothelial cells and adipogenic progenitors in $Rank^{cre/+};Csf1r^{fl/-}$ mice compared to control mice. However, endothelial cell numbers were found reduced at 3 weeks of age but all niche cells recovered to wild-type levels at 3 months of age. We conclude that during the critical time window of HSC immigration into the bone marrow, the most prevalent impact of the lack of embryonic myeloid cells is on the formation of mesenchymal lineage niche space.

The homing and retention of HSCs in the bone marrow of adult mice relies on CXCL12-CXCR4 chemokine signaling[32]. Key producers of CXCL12 in adult mice are non-hematopoietic niche cells, including mesenchymal stromal cells (MSC)[53,54], CXCL12-abundant reticular cells (CAR)[55,56] and, at lower levels, endothelial cells[54]. Cell type specific depletion of *Cxcl12* results in reduced but fully functional HSC numbers in the bone marrow of adult mice[53]. CXCL12 is also a key factor for the attraction of HSC in neonate mice[57], and is produced by nestin positive[11] and leptin receptor positive[13] cells in neonate animals. We show here that endothelial cells, osteoprogenitors and MSCs express *Cxcl12* in the newborn niche, an expression pattern that is changed already 18 days after birth, after which CXCL12 protein expression is largely confined to CD51 MSCs. The number of CD51 positive MSCs is severely reduced in newborn *Rank*^cre/+;*Csf1r*^fl/- mice in the absence of embryonic myeloid cells in vivo, and, furthermore, the amount of CXCL12 produced by remaining MSCs is decreased. In vitro, embryonic but not adult-macrophages or other myeloid cells support MSC growth pointing toward embryonic macrophages as key mediators of MSC generation also licensing MSCs for the production of large amounts of CXCL12. In contrast, in newborn mice, the numbers and function of other niche cells, including endothelial cells and adipogenic precursors is unaltered in vivo. Depletion of CXCL12 production from MSCs, but not from endothelial cells or embryonic myeloid cells results in reduced HSC numbers in the bone marrow and an accumulation of HSCs in the liver, mirroring the complete phenotype of mice devoid of embryonic myeloid cells. Taken together, our data points towards a dual role for embryonic myeloid cells in the establishment of a functional bone marrow niche space: First, the generation of MSCs, and second, their licensing to produce CXCL12.

The prevalent concept of HSC generation starts with their specification at intraembryonic sites, followed by extensive expansion in the fetal liver and migration to the bone marrow around birth, from where they replenish mature blood cells throughout life[4,5]. However, not the fetal liver[8], but the bone marrow in neonate mice[9] was suggested to support stem cell expansion, emphasizing the importance of how the bone marrow niche becomes amenable to hosting HSCs. In the absence of embryonic myeloid cells, we detect reduced numbers of HSCs in the bone marrow up to three weeks after birth despite an increased cycling activity at this time. Instead, elevated numbers of HSCs circulate through the blood and migrate to the spleen, suggesting lack of suitable niche space in the marrow. Decreased cell surface expression of CXCR4 on HSCs that migrate to the bone marrow suggests an outside-in communication between a defective niche space and incoming HSCs that as a consequence display transcriptional alterations at three weeks of age compared to controls enriched for terms associated with cell adhesion, pointing out altered migration potential. The hypothesis of a non-supportive niche impinging transcriptional alterations onto HSCs that make them less likely to stay in the bone marrow is further strengthened by reduced homing activity of transplanted hematopoietic stem and progenitor cells from *Rank*^cre/+;*Csf1r*^fl/- mice into wildtype recipients compared to controls.

Tissue-resident macrophages play important roles for homeostasis of tissues in which they reside[17,18]. We have previously contributed to this understanding through investigating the dependency of HSC-derived immune cells on specialized embryonic red-pulp macrophages in the spleen during development but also throughout life[20]. Further, we have shown that embryo-derived osteoclasts are pivotal for normal tissue architecture of the bone and that their rejuvenation depends on adult-derived monocytic cells with direct implications for the treatment of human osteopetrotic diseases[19]. We here identify a hitherto unknown embryonic macrophage population in the bone marrow that, together with embryonic osteoclasts constitute embryonic contributions to the bone microenvironment and are crucial for the formation of the stem cell niche in situ. While we formally cannot distinguish between the contribution of embryo-derived bone marrow resident osteoclasts and macrophages to the establishment of the niche, we use genetic tools and provide functional evidence ex vivo to show that embryonic myeloid cells orchestrate the non-hematopoietic niche space important for the normal founding of the hematopoietic stem cell pool. In a back-to-back manuscript[58], the authors use a complementary approach by combining genetic tools with elegant in situ imaging and conclude that yolk-sac-derived myeloid cells regulate colonization of the fetal bone marrow by definitive hematopoietic stem cells. Together, these findings significantly expand our current knowledge on niche formation in situ and support the concept that embryonic myeloid cells specify hematopoietic niche space during development.

## Methods
### Mice
All animal experiments (Mus musculus) were performed in accordance with German animal welfare legislation and were approved by the relevant authorities: Landesdirektion Dresden and the Thüringer Landesamt für Verbraucherschutz (TLV). All mice were bred and kept under specific pathogen-free conditions in separated ventilated cages in the animal facility of the TU Dresden or Leibniz Institute on Aging, Jena providing a 12 h/12 h light/dark cycle (7am-7pm) at a temperature of 22 ± 2dC and 55 ± 10% humidity (40–75% tolerance limit). They were kindly provided by: *Csf1r*[59] Richard Stanley, *Csf1r*^fl[60] Jeffrey Pollard, *Vav*^cre[61] Thomas Graf, *Cxcl12*[62], *Cxcl12-DsRed*^ki[54] Sean Morrison, *Cdh5*^cre[63] Luisa Iruela-Arispe, *Rosa26.tdRfp*^fl[64] Hans-Jörg Fehling. *Rank*^cre mice were generated by our lab[20]. C57BL/6 J (B6, #000664), *Rosa26.LSL-Yfp*^fl (#006148), *Rosa26.LSL-tdTomato*^fl (#007909), *Lepr*^cre (#032457) and *Prx*^cre (#005584) mice were purchased from the Jackson Laboratory. *Csf1r* mice were kept on C3H/HeJ, and all other mouse lines on the C57BL/6 J genetic background. Mice were sacrificed by decapitation (newborn mice) or cervical dislocation (adult mice). *Rank*^cre/+;*Csf1r*^fl/- mice were generated by first crossing *Rank*^cre/+ males with *Csf1r*^+/- females. Male *Rank*^cre/+;*Csf1r*^+/- offspring was then bred with *Csf1r*^fl/fl females to obtain *Rank*^cre/+;*Csf1r*^fl/- mice. *Vav*^cre/+;*Csf1r*^fl/- mice were generated following the same breeding strategy. *Prx*^cre/+;*Cxcl12*^fl/fl, *Cdh5*^cre/+;*Cxcl12*^fl/fl, and *Rank*^cre/+;*Cxcl12*^fl/fl mice were produced by crossing ^cre/+ males (specific to each line) with *Cxcl12*^fl/fl females. Male ^cre/+;*Cxcl12*^fl/+ offsprings were crossed with *Cxcl12*^fl/fl females. Lineage tracers were generated by crossing ^cre/+ males with *Rosa26.LSL-YFP*^fl/fl, *Rosa26.tdRFP*^fl/fl, or *Rosa26.LSL-tdTomato*^fl/fl females. Genotyping information is provided in Supplementary Data 6.

**Bulk RNA sequencing** used in Fig. 1c, Supplementary Fig. 1c: modified from[65] 80–90 **HSCs** (KSL CD48- CD150+, 3 week-old mice) were FACsorted into 2 ul 0.2% Triton-X 100 supplemented with 4 U murine RNase Inhibitor (NEB, M0314L) and frozen at -80 °C. After thawing the samples, RNA is denatured for 3 min at 72 °C in the presence of 2.4 mM dNTP (Invitrogen, R0192), 240 nM dT-primer* and 4 U RNase inhibitor (NEB). The reverse transcription and addition of the template switch oligo was performed at 42 °C for 90 min after filling up to 10 μl with RT buffer mix for a final concentration of 1× superscript II buffer (Invitrogen), 1 M betaine, 5 mM DTT, 6 mM MgCl₂, 1 μM TSO-primer, 9 U RNase Inhibitor and 90 U Superscript II, before heat-inactivation at 70 °C for 15 min. With 10% of the material, a qPCR on full-length cDNA was performed with universal primers to determine the optimal number of cycles to avoid under- or overamplification of the samples. The single stranded cDNA was subsequently amplified using Kapa HiFi HotStart Readymix (Roche, 07958935001) at a 1× concentration together with 250 nM UP-primer under following cycling conditions: initial denaturation at 98 °C for 3 min, 17–22 cycles [98 °C 20 sec, 67 °C 15 sec, 72 °C 6 min] and final elongation at 72 °C for 5 min, followed by a cleanup with 1× volume of Sera-Mag SpeedBeads (GE Healthcare, 65152105050250) resuspended in a buffer consisting of 10 mM Tris, 20 mM EDTA, 18.5% (w/v) PEG 8000 and 2 M sodium chloride solution. For library preparation, 2 μl cDNA was tagmented at

55 °C for 15 min in a total volume of 4 µl containing 1× Tagment DNA Buffer and 0.8 µl Tagment DNA Enzyme (from the Illumina DNA Prep− Tagmentation Kit, Illumina, 20060059). The reaction was stopped with 0.02% SDS at 37 °C for 15 min. After removing the supernatant, the index PCR run under following conditions: (72 °C 3 min, 98 °C 30 sec, 12 cycles [98 °C 10 sec, 63 °C 20 sec, 72 °C 1 min], 72 °C 5 min) in 1× KAPA HiFi HotStart Ready Mix and 0.7 µM dual indexing primers. The libraries of one *Rank*[cre/+]*;Csf1r*[fl/−] and control sample was prepared like described above (method 1). The library preparation of one other *Rank*[cre/+]*;Csf1r*[fl/−] and 2 control samples was processed as follows (method 2): 3 ng cDNA was tagmented at 55 °C for 5 min in a total volume of 5 µl containing 1× TruePrep Tagment Buffer L, 0.5 ul True-Prep Tagment Enzyme V50 (from TruePrep DNA Library Prep Kit V2 for Illumina; Vazyme, TD501-02). Subsequently, Illumina indices are added during PCR (72 °C 3 min, 98 °C 30 sec, 12 cycles [98 °C 10 sec, 63 °C 20 sec, 72 °C 1 min], 72 °C 5 min) with 1× concentrated KAPA Hifi Hot-Start Ready Mix and 300 nM dual indexing primers.

After PCR, all libraries were purified with 0.9x volume of rebuffered Sera-Mag SpeedBeads (GE Healthcare), followed by a size selection with 0.6x (right side) and 0.9x (left side) volume of Sera-Mag SpeedBeads. The libraries were quantified with the Fragment Analyzer, and sequenced with a Novaseq 6000 system (Illumina) on a S4 flowcell in 100 bp paired-end XP mode, aiming an average sequencing depth of 45 million fragments per library.

Fig. 3g, Supplementary Fig. 3c–f: Bulk sequencing of **embryonic eYFP+ and adult eYFP- bone marrow macrophages**: 3618–4000 bone marrow myeloid cells (F4/80[hi], Ly6C[hi] monocytes, Ly6C[lo] monocytes, PMNs) from 8-week-old mice were sorted into QIAzol Lysis Reagent (Qiagen, #79306). RNA isolation was performed with miR-Neasy Micro Kit (Qiagen, #217084). After RNA isolation, the volume was evaporated to 2 µl (with a vacuum centrifuge for 12 min at room temperature). and processed similar to the HSCs samples with 14–18 cycles amplification of the full-length cDNA and identical library preparation protocol (method 1). The libraries were sequenced with a Novaseq 6000 system (Illumina), aiming at a minimum sequencing depth of 35 mio. fragments per library.

Supplementary Fig. 4c: 100 adult **MSCs** (CD45[-] Ter119[-] CD31[-] CD51[+] PDGFRa[+] Sca1[-], 14-week-old mice) were FACsorted into 2 ul 0.2% Triton-X 100 supplemented with 4 U murine RNase Inhibitor (NEB) and frozen at -80 °C. The samples were processed following method 1.

## Isolation of hematopoietic and niche cells

Embryos and newborn mice: timed pregnancies were performed and the day of vaginal plug was evaluated as 0.5 days post-conception. Aorta-Gonad-Mesonephros (AGM) and fetal livers from E10.5 embryos were digested 30 min at 37 °C in PBS/5% FCS containing collagenase type 4 (Worthington, 100 µg/ml final,#CAS:9001-12-1) and DNAse I (Sigma-Aldrich, #DN25-1G, 100 µg/ml final). The reaction was stopped by adding 12.5 mM EDTA. Fetal spleen and bone marrow, and newborn liver, spleen, and bone marrow were disintegrated between the frosted end of two glass slides without digestion. All samples were filtered through a 100µm filter mesh. Niche cell preparation for newborn, 3-week and 14-week-old mice: Femurs were prepared[39]. Briefly, femurs were flushed twice with PBS and digested for 30 min at 37 °C under agitation. Digestion was stopped with PBS/5% FCS, the suspensions were filtered through a 40 µm filter mesh and continued with staining or differentiation culture. Hematopoietic cell isolation from 3-week-old mice: Bones were crushed, and spleens gently disintegrated between the frosted end of two glass slides. Red blood cell lysis (ACK Lysing Buffer, Gibco #A10492-01) was performed for bone marrow (20 sec) and spleen (40 sec). Lysis was stopped by adding PBS/5% FCS. Blood samples: Blood samples were obtained by retro-orbital bleeding. Red blood cell lysis was performed twice for 5 min each. Lysis was stopped by adding PBS/5% FCS.

## Flow cytometry

Cells were stained and blocked with purified CD16/32 (Invitrogen, clone 93) and rat Immunoglobulin (Jackson Immuno Research, 012-000-002) for 40 min on ice[66]. Macrophage samples are blocked with purified CD16/32 and rat Immunoglobulin) for 30 min on ice before staining. All antibodies are listed in Supplementary Data 5. Gating strategies are shown in Supplementary Fig. 5. Counting beads (CountBright, ThermoFischer, #C36950) were added to each sample during staining. Live/dead discrimination is done by 0.4 µg/ml of DAPI (Biochemica, A1001,000) right before acquisition.

For sorting, macrophages were enriched using biotinylated anti-F4/80 antibodies and then further incubated with anti-biotin beads (Miltenyi Biotec #130-090-485). HSC and HSPC were sorted from the lineage (CD45, Ter119, CD3, CD4, CD8, Gr1, CD11b)-depleted bone marrow cells (MACS). Before acquisition cells were passed through a 40 µm filter mesh. Cell cycle: Bone marrow and spleen cells were stained for cell surface antigens, fixed using BD Cytofix/Cytoperm Fixation and Permeabilization solution (BD #51-2090KZ), frozen (−80 °C) and subsequently stained. Samples were acquired and sorted using BD LSRII, BD LSRFortessa, BD FACSAria III, BD FACSAria Fusion and analyzed with FlowJo Software (TreeStar).

## Transplantation

Figure 1f: 1000 bone marrow KSL cells (CD45.2) from *Rank*[cre/+]*;Csf1r*[fl/−] or littermate wildtype controls (21–25-day-old) were sorter purified and transplanted into adult RgW41 (CD45.1) recipients[34]. Bone marrow chimerism was evaluated 32 weeks after transplantation.

Figure 2f: $4 \times 10^5$ whole bone marrow cells from newborn *Rank*[cre/+]*;Csf1r*[fl/−] or littermate controls (2-day-old) were transplanted intravenously into adult RgW41 recipients. Blood PMN chimerism was assessed at 5, 9, 14 and 18 weeks and bone marrow HSC (KSL CD48-CD150 + ) chimerism was evaluated 18 weeks after transplantation. Fig. 3d 1000 bone marrow KSL cells (CD45.2) from wildtype mice (21–25-day-old) were sorter purified and transplanted into adult RgW41 (CD45.1) recipients. Bone marrow chimerism was evaluated 32 weeks after transplantation.

## In vivo Homing assay

4x10E5 bone marrow cells from newborn *Rank*[cre/+]*;Csf1r*[fl/−] or littermate controls were transplanted into the liver of neonate wild-type mice. 16 h later, bone marrow samples were stained and analyzed.

## In vitro Migration assay

750–800 liver KSL cells from newborn *Rank*[cre/+]*;Csf1r*[fl/−] or littermate wildtype (WT) control mice were sorter purified and dispensed to the upper part of the 24-well transwell plate (VWR 8.0 µm, 24 well) containing DMEM (Gibco, 31966-021) with 10% FCS and 1% P/S. The bottom part of each transwell contained 100 ng/ml of murine CXCL12 (rm-SDF1 Peprotech, #250-20A-10UG) or not. After 2.5 h of incubation at 37 °C, cells from the bottom were collected and analyzed by flow cytometry.

**Bioinformatics.** Fig. 1c, Supplementary Fig. 1c: FastQC (http://www.bioinformatics.babraham.ac.uk/) was used to get basic quality reports of the sequencing data. The quality of fragments was assessed by examining the mean quality scores, gc content, adapter content and per sequence quality scores. Reads were aligned to the mouse reference (mm10) with support of the Ensembl 98 splice sites using the aligner gsnap (v2020-12-16)[67]. Fragments per gene and sample were obtained based on the overlap of the uniquely mapped fragments with the same Ensembl gene annotation using featureCounts (v2.0.1)[68]. DESeq2 R package (v1.30.1)[69] and Independent Hypothesis Weighting (v1.18.0)[69] were used for following exploratory analysis and differential gene expression analysis. Genes with a maximum of 10% false

discovery rate (padj ≤ 0.1) and log2 fold change of < −2 and > 2 were considered as significantly differentially expressed.

Figure 3g, Supplementary Figs. 3c–e, 4c: Our in-house RNA-seq analysis pipeline rippchen v0.10.0[70] was applied to calculate differentially expressed genes and to perform gene set enrichment tests. Therefore, the mouse reference genome GRCm39 (mm11) and its annotation were obtained from Ensembl v109 and raw sequencing data was converted into paired-end reads using bc2fastq v2.20.0.422. According to FastQC v0.11.9 reports inferred Nextera Transposase adapter contamination, Trimmomatic[71] v0.39 was utilized to perform quality trimming (5nt sliding window, mean quality cutoff 20) and Cutadapt[72] v2.10 to perform removal of 3' adapter as well as poly mono-nucleotide content. Rcorrector[73] v1.0.4 derived, sequencing-error-corrected data was subsequently filtered by SortMeRNA[74] v2.1 for ribosomal RNA fragments. The preprocessed data was then aligned to the indexed reference genome using segemehl[75,76] v0.3.4 with adjusted accuracy cutoff (95%) and enabled to map split reads (-S). Unambiguously aligned read pairs were quantified via featureCounts[68] v2.0.1 on exon meta-features, with a minimum overlap of 10nt. Significantly differentially expressed genes across all conditions were estimated using DESeq2[69] v1.34.0 and a Benjamini-Hochberg method adjusted p-value cutoff of 0.05. Over-representation tests (background genes selected by transcript per million (TPM) >= 1), gene set enrichment tests for protein-coding genes, ranked according to adaptive shrunken fold-changes, respectively, were conducted for gene sets obtained from MSigDB[77] v2023.2.Mm or Ensembl v109, utilizing clusterProfiler[78] v4.2.0 (10 <= set-size <= 500). Enriched gene sets were semantically clustered using rrvgo[79] (Wang's method, cutoff 0.7).

## Lysotracker

Bone marrow samples were incubated with 50 nM Lysotracker (Invitrogen #DND-22) for 40 min at 37 °C. Cells were washed with PBS/5% FCS and stained. To identify dead cells, a fixable viability stain was added (BD #564997).

## Phagocytosis

Bone marrow cells were incubated with 100 µg/ml pHrodo™ E.coli BioParticles (ThermoFischer, #P35361) in RPMI 1640 Medium supplemented with Glutamax (Thermo Fischer Scientific, #61870036) containing 10% FCS at 37 °C. Every 15 min, the cells were gently pipetted up and down to avoid adherence.

## Immunofluorescence

Femurs from newborn *Rank^cre/+;Csf1r^fl/−* or control mice incubated with 4% PFA overnight at 4 °C. Bones were washed, dehydrated, and embedded in paraffin. Bones were sectioned 5 µm longitudinally with a microtome (Thermo Fischer Scientific HM 340E Rotary Microtom). For stainings, bones were deparaffinized and antigen retrieval was applied for 5 min at 330 Watt (microwave) in a glass chamber with EDTA Buffer (40 mM Tris pH:8.0 / 1.2 mM EDTA pH:8.0/water). Bone sections were rinsed with PBS and incubated in blocking buffer (3% BSA with 0.3% Triton X-100 in PBS, 1 h RT) and subsequently stained overnight at 4 °C in the dark. Secondary steps were incubated for 1-2 hrs at RT in the dark. Slides were washed 3 times and DAPI was added for 4 min (2 µg/ml, BioChemica, #A1001,000). Slides were washed 3 times with PBS for 5 min each time and then were mounted (DAKO, #S3023). Pictures were taken by BZ-9000E (Keyence) and images were processed by Fiji Software. From 3 different vision fields per bone/mouse, IntDen, Area, and Mean Fluorescence values were obtained from the selected positively or negatively labeled cells, and Corrected total cell fluorescence (CTCF) was calculated. CTCF = IntDen-(Area of selected cell x (mean fluorescence of the negative cells). The mean CTCF value per bone was calculated and plotted in the graphs.

## Metatarsal Assay

The metatarsal assay was performed as described[80]. Briefly, metatarsal bones were gently removed and placed in a 24-well plate (Cell Star) pre-coated with 0.1% gelatin (Gelatin Solution, type B, 2%, Sigma-Aldrich #G1393-100 ml) in 250 µl water. Drop by drop, 250 µl of a-MEM (Gibo, 22561-021) supplemented with 10%FCS and 1% P/S were added to wells. Every 48 h, the medium was exchanged. On day 12, the medium was removed, and wells were washed with D-PBS and fixed with 4% PFA for 20 min. PFA was removed and wells were washed twice with D-PBS. Samples were blocked by adding 200 µl of Blocking Buffer consisting of (3% TritonX-100, 1% Tween-20, and 0.5% BSA) with gentle rocking for 1 h at room temperature. Wells were incubated with polyclonal goat CD31 (R&D #AF3628) antibody at room temperature for 2 h, washed twice with a blocking buffer and wells were incubated with secondary antibody donkey anti-goat IgG (H+L) conjugated with A488 (Thermo Fisher Scientific #A-11055) overnight at 4 °C. The following day, wells were washed twice with a blocking buffer and imaged with microscope BZ-9000E (Keyence). Images were processed by Fiji Software. Pixel area measurement from vascularized and bone area was calculated and vascularized area was divided by the bone area to show the proportion of vascularization relative to bone size. The mean value of the vascularized/bone area was plotted per individual mice.

## Osteogenic Differentiation

Osteogenic differentiation was conducted as described previously[81]. Briefly, 2.50E + 05 newborn bone marrow cells were plated in 300 µl Osteogenic Differentiation Medium that contains ascorbic acid-free a-MEM (Gibco, A1049001), 10% FCS, 1% P/S, supplemented with β-Glycerophosphate disodium salt hydrate (BGP Sigma-Aldrich, G9422-10G) and 2-phospho-L-ascorbic acid trisodium salt (2P-AA final 50 µg/ml Sigma 49752-10 G) for 11 days in 48 well plates (Cell Star). Medium was replaced every third day. On day 11, wells were washed with PBS. Cells were fixed with 4% PFA for 30 min, washed with PBS, and stained with 300 µl of Alizarin Red Solution (Merck #2003999) for 30 min at room temperature. Wells were washed and images were taken by BZ-9000E (Keyence). Later, 600 µl of destaining solution (70% water, 10% acetic acid, and 20% methanol) was added to the wells and incubated for 30 min at room temperature. Later, 300 µl destaining solutions per sample is added to the Cell Star 96 well plate F-bottom for micro-plate reader TECAN Infinite M1000 Pro. For alizarin red absorbance measurement, 450 nm wavelength was selected.

## Adipogenic Differentiation

Adipogenic differentiation was conducted as described previously[81]. Briefly, 2.50E + 05 newborn bone marrow cells were plated in 200 µl a-MEM (Gibco, 22561-021) with 10% FCS and 1%P/S. The following day, the medium was exchanged with adipogenic medium containing a-MEM, 10%FCS, 1%P/S, 100 nM Dexamethasone (D2915-100mg, Sigma Aldrich), 10 µg/ml Insulin (Sigma Aldrich I5500-100 mg), 125 µM Indomethacin (Sigma Aldrich I7378), 0.5 mM IBMX (Sigma Aldrich I5879) and 1 nM triiodothyronine (T3, stock: 1 µM, Sigma Aldrich T2877) and cultures kept for 48 h. At day 3, the adipogenic medium was replaced with a-MEM, 10% FCS, 1% P/S, 100 nM Dexamethasone, 10 µg/ml Insulin. On day 5, the medium was exchanged. On day 7, cells were washed with PBS, and cells fixed with 200 µl 4% PFA for 30 min RT. PFA was removed and wells were washed with 200 µl PBS and 200 µl 60% isopropanol solution and stained immediately. 100 µl of staining: 3 parts of freshly filtered (0.22 µm) Oil red O solution (Stock:0.3% Oil red O in 100% isopropanol, Sigma Aldrich, O0625-25G) with 2 parts of water added to the wells and incubated 30 min at room temperature. Wells were washed with PBS (200 µl) and images were taken by BZ-9000E (Keyence). Next, 200 µl of destaining solution (100% isopropanol) was added to the wells and incubated for 30 min. 200 µl destaining solutions per sample is

added to the Cell Star 96 well plate F-bottom for micro-plate reader TECAN Infinite M1000 Pro. For Oil red O absorbance measurement, 510 nm wavelength was selected.

## Co-culture myeloid cells and MSCs

Figure 3j: 1000 MSCs from bones of newborn mice (1–5 days, 5–8 mice per experiment) were sorter purified and cultured with or without sorter-purified bone marrow myeloid cells (1000 cells, macrophages, monocytes, and PMNs) from 19- to 21-days-old mice (2 per experiment) in a 96-well flat bottom plate (Cell Star) in osteogenic differentiation medium. 8 days later, MSCs were stained and numbers were calculated by flow cytometer. Three independent experiments were conducted.

Supplementary Fig. 4b: For osteogenic differentiation after co-culture, 5000 MSCs from bones of wild-type mice (6-8-day-old) were sorter purified and cultivated with or without 5000 sorter-purified eYFP+ or eYFP- macrophages from 3–11-week-old mice in a 96-well flat bottom plate (Cell Star) in osteogenic differentiation medium. 50% of the medium was exchanged every fourth day (containing 2× supplements). 14 days later, wells were stained with alizarin red staining and absorbance was measured.

## Fold-change calculations

Fold changes were calculated by dividing each data point of mutant and control mice with the mean of wild-type samples for each individual experiment.

## Statistics

Student's unpaired, two-sided t-test was performed to calculate the statistical significance between individual groups with expected normal distribution. Mann–Whitney U test was performed to calculate the statistical significance between groups where normal distribution was not expected (fold-change comparisons). $P$ value $(P) > 0.05$; $*P \leq 0.05$; $**P \leq 0.01$; $***P \leq 0.001$ and $***P \leq 0.0001$). Prism 9 software (GraphPad) was used to perform statistical analysis. If not indicated differently, the mean and individual data points are shown.

## Reporting summary

Further information on research design is available in the Nature Portfolio Reporting Summary linked to this article.

## Data availability

Sequencing and processed data generated in this study have been deposited on Gene Expression Omnibus (GEO) under accession numbers GSE265965 for bulk RNA-seq data of sorted HSCs from $Rank^{cre/+};Csf1r^{fl/-}$ and control BM, GSE265827 for bulk RNA-seq data of murine bone marrow macrophages of embryonic and adult origin, GSE282288 for bulk RNA-seq data of niche cells. Full descriptions of experimental procedures and bioinformatic methods can be found in the Method section. All data generated or analyzed during this study are included in this published article and its Supplementary Information files or from the corresponding author upon request. Source data are provided with this paper.

## Code availability

KR, SH: Source code is deposited here: https://github.com/hoffmann-lab/rippchen and was used in version 0.10.0[70]. ML: Source code is deposited here: https://github.com/mlescheDCGC/deseq2_publication/releases/tag/v1.0 or https://doi.org/10.5281/zenodo.14870954[82]. JM, TH: Commands used to generate the count matrix of MSCs: https://github.com/hoefer-lab/Xpand_bulk_RNA_raw_data_processing or https://doi.org/10.5281/zenodo.15017377[83].

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

## Acknowledgements

The authors thank Sabrina Eichwald for excellent support in histological analysis. The authors further thank the FLI histology facility for expert bone cuts, and the animal houses of the FLI and the TU Dresden for mouse husbandry. This work was supported by the German Research Foundation (DFG) through CRC1278 (project number 316213987, project C06), WA2837/7-1, WA2837/8-1, by the Carl Zeiss Foundation (Impuls, project ID P2019-01-006), and by the DFG under Germany´s Excellence Strategy – EXC 2051 – Project-ID 390713860 (all C.W.). The project was further supported by the Leibniz Association (XpandHSC, project ID K243/2019 to C.W. and T.H.).

## Author contributions

G.P. designed the study, performed, analyzed and interpreted data and wrote the manuscript. C.W. conceived and designed the study, supervised experiments, analyzed and interpreted data and wrote the manuscript. S.R. performed sequencing, K.R., M.L., J.M., T.H. and S.H. performed bioinformatic analyses. J.F. performed MSC cell sort for sequencing. S.C. performed RNA isolation and handled animal experiment licensing.

## Funding

## Competing interests

The authors declare no competing interests.
