## [Transparent Peer Review file · Nature Communications]

Embryonic macrophages orchestrate niche cell homeostasis critical for the establishment of the definitive HSC pool.

Corresponding Author: Professor Claudia Waskow

Version 0:

Reviewer comments:

Reviewer #1

(Remarks to the Author)

The role of embryonic macrophages in regulation of hematopoietic stem cells (HSCs) and their niches in bone marrow remain unclear. The authors show that bone marrow macrophages consist of two ontogenetically distinct cell populations arising from yolk sac and definitive HSCs and a proportion of yolk sac-derived macrophages declines over time and that the numbers of macrophages were markedly reduced in the bone marrow of Rank-cre; CSF1 receptor (Csf1r) fl/- mice, which lacked embryonic macrophages, at 3 weeks of age. In addition, they show that in Rank-cre; Csf1r fl/- mice, the numbers of HSCs were decreased in the bone marrow at 3 weeks of age but increased in liver at 0-4 days of age although HSC numbers were unaltered in 3-week old Vav-cre; Csf1r fl/- mice, which lacked macrophages generating from definitive HSCs. This is an interesting, important, and well-written study. Nevertheless there are some concerns that need to be addressed.

1. Due to a lack of genetic studies, the role of macrophages in regulation of HSCs and hematopoiesis remains incompletely understood. Thus, it is important to show the numbers of HSCs and hematopoietic progenitors, including common lymphoid progenitors (CLPs), pre-B cells, granulocyte/macrophage progenitors (GMPs), granulocytes, megakaryocyte/erythrocyte progenitors (MEPs), and proerythroblasts in adult and aged control, Rank-cre; Csf1r fl/-, Csf1r -/-, and Vav-cre; Csf1r fl/- mice.
2. Figure 3f: The authors should show the numbers of bone marrow macrophages in 3-week old and adult Vav-cre; Csf1r fl/- mice.
3. Since it is important to know the role of ontogenetically distinct cell populations of macrophages in the formation of the bone marrow cavity, the authors should show bone marrow volume of femurs in E18.5, newborn, and 1-week old control, Rank-cre; Csf1r fl/-, and Csf1r -/ mice.
4. It is generally assumed that macrophages regulate erythropoiesis in the bone marrow. To clarify the extent of the contribution, I would recommend the authors to show erythropoiesis, such as the numbers of erythroblasts and enucleated reticulocytes in the marrow and red blood cell numbers in the peripheral blood in adult control, Rank-cre; Csf1r fl/-, Csf1r -/-, and Vav-cre; Csf1r fl/- mice.
5. Figure 4: Flow cytometric analysis showed that the numbers of endothelial cells and CD51+PDGFRa+ cells were severely reduced in newborn Rank-cre; Csf1r fl/- mice. This result raises the possibility that major function of yolk sac-derived macrophages is to form bone marrow cavity and/or to regulate vascular formation in the marrow, resulting in retarded invasion and/or differentiation of HSC niche cells. To address this possibility, the authors have to show the numbers of CD31+ endothelial cells and PDGFRb+ HSC niche cells in the bone marrow of Rank-cre; Csf1r fl/- mice at the age of 1 week and 3 weeks by histological and flow cytometric analyses.

Minor points

1. Page 8, line 233: myeloid cells should be mesenchymal cells.

2. Page 10, line 270: Since mononuclear cells would include mesenchymal cells, mononuclear cells should be macrophages or macrophage-lineage cells.

3. Page 11, line 306 and 322: The authors described that endothelial cells were unaltered in numbers. However, endothelial cell numbers appear to be severely reduced by histological and flow cytometric analyses (Figure 4a-c).

4. Page 11, line 314: Nestin is expressed in endothelial cells but not in CXCL12-expressing HSC niche cells in the mouse bone marrow (Zhou et al., Cell Stem Cell 15; 154, 2014; Baryawno et al., Cell 177; 1, 2019).

Reviewer #2

(Remarks to the Author)

The authors present a large body of work analyzing the role of yolk sac derived macrophages in sculpting the definitive HSC niche in the bone marrow. They use several conditional knock out mice and identify a number of interesting phenotypes in mice that lack embryonic but not bone marrow derived macrophages. They conclude that in early development yolk sac derived macrophages are important to regulate bone development and MSCs, with an important effect on the levels of CXCL12 production by the niche and sensing by HSCs. However, a number of conclusions remain overstated and need to be more deeply substantiated, and a number of experimental models and approaches need to be better clarified and justified. See details below.

Methods are referenced but missing. They must be included for the manuscript to be evaluated.

Why are *csf1r^{-fl}* used rather than *fl/fl*? If the rank cre is not efficient at deleting a double floxed allele, this should be shown. The mice studied are in effect already hemizygous and given the importance of embryo-derived myeloid cells, a number of the effects seen may be due to this.

Are HSC numbers normal or normalising in *csf1r* rank cre mice older than the ones shown? It is important to show adult mice, especially because the higher cell cycle observed in young ones may lead to overall normal adult hematopoiesis. How about *csf1r* ko HSCs? The same assays should be shown as for the rank cre *csf1r^{-fl}* mice. Should differences between the two strains be expected and are there any?

HSC numbers in spleen and liver over time - do they balance?

Was the transcriptomic analysis also performed at 3 weeks of age? HSC *Cxcr4* expression reduced in scRNAseq from adult? This finding needs to be followed up. In fact, every transcriptomics analysis needs follow up, as on its own has little value.

Why would *Cxcr4* be downregulated? Could it be due to *cxcl12* levels being diminished? This needs proving using the appropriate mouse crossings.

Rank cre deletes from embryonic development, but is there proof that it does not delete in adult macrophages and/or osteoclasts?

Figure 3b-d work needs to be better explained and requires a deeper characterization of the mouse strains used.

Figure 3h-l. It is unclear how this was performed and what is the reason to present these data.

What cells are MSCs? Nowhere it is explained what phenotypic or functional markers are used to purify these cells. Do they overlap with *LepR* positive cells?

Fig 4 shows consequences on osteoblasts not MSCs, as *CD51* expression is linked to osteoblast differentiation. Of note, *CD51* is indicated as a myeloid marker in the discussion section. *CD31* is greatly diminished, indicating a vascular phenotype is present too. Rewording of this results section is needed, and the findings need to be put within the context of known role of osteoclasts led bone remodeling in bone development in discussion.

Are the ko mice smaller? Are the bones smaller/thinner/less trabecular?

Would co-culture with macrophages restore osteogenicity of MSCs ?

Findings using *prx* cre are correlative not mechanistic. Are *prx* cre *r26* YFP cells diminished in rank cre *csf1r* ko mice?

Discussion needs to be very clear about what age is being affected in what way.

Rank would affect osteoclasts so not possible to discriminate between macrophages and osteoclasts

Reviewer #3

(Remarks to the Author)

This is a very nice study that uses elegant mouse genetics to dissect the role of embryonic macrophages in establishment of definitive HSCs within the BM. The authors demonstrate a loss of BM HSCs in postnatal life associated with impaired fetal myeloid development through EMP-specific deletion of *Csf1r1*. The Rank-Cre system has been shown in a previously publication to definitively label and delete EMP-derived osteoclasts. The authors demonstrate that the effect on definitive HSCs is not due to an effect on HSC emergence or of deletion of *csf1r* on HSCs or other hematopoietic cells (with *Vav-Cre*), and also demonstrate that there is a compensatory increase in HSCs in the blood and spleen, suggestive of a homing defect. HSCs in mutant mice show downregulated *CXCR4* and impaired homing. Finally, deletion of EMP-derived myeloid cells results in impaired MSC and osteoblast precursor development, and a reduction in *CXCL12* production from MSCs, which the authors assert as a mechanisms for impaired HSC homing and seeding in the BM. The authors use additional genetic mouse models to dispel a mechanism whereby MSC-derived *CXCL12* production recapitulates the phenotype of

EMP-derived myeloid cell deletion.

Overall, this is a very thorough study in which the data generally provide strong support for the conclusions. The findings are also quite novel and of general interest to the hematopoiesis community. Nonetheless, there are a few specific areas where the manuscript could potentially be improved:

In general, the authors use their Rank-Cre X Csf1r^{fl}/- model to delete “fetal-derived myeloid cells” in the bone marrow. Not until the very end of the manuscript do they mention osteoclasts. The reason for this omission is unclear to the reviewer. I think it would be beneficial to the manuscript to state/address this earlier, perhaps when discussing the identification of the F4/80 population in Fig. 3.

In Figure 1F, the authors state that they transplant BM HSC (line 132), but they are actually transplanting LSK. These are not the same, and there is a difference in the frequency of HSCs within LSK between WT and mutant. This should be justified and/or addressed within the results and discussion.

Given this difference, the differences in outcomes between 1F and 2F are somewhat confusing. In 1F, mutant LSK should have less HSCs, but they appear to have more HSCs in the BM at 32 weeks. Why would transfer of WBM in 2F yield such different results as compared to LSK?

In Fig 2H – Did cells that did not go to the BM home to spleen and PB instead?

In Figure 3, the RNA seq data would benefit from improved data presentation and some further analysis. Data in SFig3 are a bit confusing, particularly SFig 3E. It would be beneficial if the authors could identify some of the critical downregulated genes associated with myeloid leukocyte migration that they think are drivers of the observed effects.

Given that embryonic macrophages support MSCs, is there any information in the RNA-seq dataset to also support this *in vivo* observation?

Figure 3J – presentation of data could be simplified by utilizing label on the X axis.

In Figure 4; the authors describe “Unchanged endothelial cells” – line 233 – but 4a appears to be considerably less ECs, and the conversion for “fold change” in 4b is a bit confusing and not intuitive. Can the authors provide more explanation?

Similarly, in 4C, low N makes the absence of a difference in CD31 staining difficult to discern. Adding more N to this experiment would be appropriate to confirm these results.

Is HSC function similarly unperturbed when CXCL12 is deleted from MSCs?

Version 1:

Reviewer comments:

Reviewer #1

(Remarks to the Author)

The authors have given a satisfactory response to my concerns, improving the manuscript with successfully. Since the additional data showing that the frequency of common lymphoid progenitors (CLPs) was reduced in the bone marrow of Rank-cre; Csf1r^{fl}/- mice at the age of 3 weeks is interesting, I would recommend the authors to show it in the main Figure and not in the Supplementary Figure 1a.

Reviewer #2

(Remarks to the Author)

I commend the authors for their thorough revisions, for highlighting how the present manuscript relates to and builds on their prior studies, and for adding excellent clarity to many points. All my concerns have been addressed.

Reviewer #3

(Remarks to the Author)

In this reviewed manuscript, the authors have satisfactorily addressed all previous concerns and thereby significantly improved the manuscript.

Point-by-point response to the reviewer's comments, MS NCOMMS-24-20946-T

Reviewer #1 (Remarks to the Author):

The role of embryonic macrophages in regulation of hematopoietic stem cells (HSCs) and their niches in bone marrow remain unclear. The authors show that bone marrow macrophages consist of two ontogenetically distinct cell populations arising from yolk sac and definitive HSCs and a proportion of yolk sac-derived macrophages declines over time and that the numbers of macrophages were markedly reduced in the bone marrow of *Rank^{cre/+};Csf1r^{fl/-}* mice, which lacked embryonic macrophages, at 3 weeks of age. In addition, they show that in *Rank^{cre/+};Csf1r^{fl/-}* mice, the numbers of HSCs were decreased in the bone marrow at 3 weeks of age but increased in liver at 0-4 days of age although HSC numbers were unaltered in 3-week old *Vav^{cre/+};Csf1r^{fl/-}* mice, which lacked macrophages generating from definitive HSCs. This is an interesting, important, and well-written study. Nevertheless there are some concerns that need to be addressed.

We thank the referee for the concise summary of our findings and we highly appreciate the positive comments.

1. Due to a lack of genetic studies, the role of macrophages in regulation of HSCs and hematopoiesis remains incompletely understood. Thus, it is important to show the numbers of HSCs and hematopoietic progenitors, including common lymphoid progenitors (CLPs), pre-B cells, granulocyte/macrophage progenitors (GMPs), granulocytes, megakaryocyte/erythrocyte progenitors (MEPs), and proerythroblasts in adult and aged control, *Rank-cre; Csf1r fl/-*, *Csf1r -/-*, and *Vav-cre; Csf1r fl/-* mice.

We thank the referee for this suggestion, CMP, GMP, and MEP frequencies are not altered at three weeks of age, suggesting that myeloid progenitor pool size is regulated independently of the pool size of HSCs. In contrast, CLP frequencies are reduced in mice lacking embryonic but not in mice lacking adult monocytes and macrophages. We have added the data into Supplementary Fig.1a and included the information into the body text. For the analysis of red blood cell generation we would like to refer to our answer to question 4 of this referee.

HSC numbers are recovered to wild type levels after 70 days of age and myeloid progenitor numbers remain unchanged compared to wild type controls, suggesting presence of compensatory mechanisms resulting in the expansion of the stem cell pool over time We have added the data into Supplementary Fig.1b and included the information into the body text.

2. Figure 3f: The authors should show the numbers of bone marrow macrophages in 3-week old and adult *Vav-cre; Csf1r fl/-* mice.

We agree that this is important information, and the data of 3 week old mice has been added into Fig.3f. Information on bone marrow macrophages numbers in adult *Vav^{cre/+};Csf1r^{fl/-}* and *Rank^{cre/+};Csf1r^{fl/-}* mice has been included as figure part b into Supplementary Fig.3. All new data is called-out in the body text.

3. Since it is important to know the role of ontogenetically distinct cell populations of macrophages in the formation of the bone marrow cavity, the authors should show bone marrow volume of femurs in E18.5, newborn, and 1-week old control, *Rank-cre; Csf1r fl/-*, and *Csf1r -/-* mice.

We thank the referee for this suggestion. Our team has shown before that embryonic macrophages are key for the generation of osteoclasts and that, consistently, *Rank^{cre/+};Csf1r^{fl/-}* mice are osteopetrotic (Jacome-Galarca, Percin et al., nature, doi: 10.1038/s41586-019-1105-7). We have shown that bone cavity formation is impaired in 1 week-old *Rank^{cre/+};Csf1r^{fl/-}* and *Csf1r^{-/-}* mice in Supplementary Fig.2 in the publication mentioned above. We have provided this information in the body text of the current manuscript and cited our previous publication.

4. It is generally assumed that macrophages regulate erythropoiesis in the bone marrow. To clarify the extent of the contribution, I would recommend the authors to show erythropoiesis, such as the numbers of erythroblasts and enucleated reticulocytes in the marrow and red blood cell numbers in the peripheral blood in adult control, Rank-cre; Csf1r fl/-, Csf1r -/-, and Vav-cre; Csf1r fl/- mice.

We appreciate the reviewer's suggestion and agree that this is an important question deserving further investigation. Erythroblastic island macrophages (EBI-macs) are important niche cells for red blood cell generation. To explore whether embryonic macrophages potentially contribute to the functional pool of EBI-macs, we analyzed reticulocyte and mature red blood cell counts in 3-week-old *Rank^{cre/+};Csf1r^{fl/-}* and *Vav^{cre/+};Csf1r^{fl/-}* mice and have found both reduced in *Rank^{cre/+};Csf1r^{fl/-}* but not in *Vav^{cre/+};Csf1r^{fl/-}* mice, suggesting that Rank-positive embryonic macrophages are an important component of the EBI-mac population. While this is exciting information, we feel that a thorough mechanistic analysis of this finding would go beyond the scope of the current manuscript focusing on the role of macrophages on the establishment of a proper stem cell niche.

5. Figure 4: Flow cytometric analysis showed that the numbers of endothelial cells and CD51+PDGFRa+ cells were severely reduced in newborn Rank-cre; Csf1r fl/- mice. This result raises the possibility that major function of yolk sac-derived macrophages is to form bone marrow cavity and/or to regulate vascular formation in the marrow, resulting in retarded invasion and/or differentiation of HSC niche cells. To address this possibility, the authors have to show the numbers of CD31+ endothelial cells and PDGFRb+ HSC niche cells in the bone marrow of Rank-cre; Csf1r fl/- mice at the age of 1 week and 3 weeks by histological and flow cytometric analyses.

To address this point experimentally, we have, in addition to the analysis in newborn mice (Fig.4a-c) performed flow cytometry of bones of 3 week- and 70-day-old mice. At 3 weeks of age, endothelial cell frequency are significantly reduced whereas MSCs and OBL are unaltered compared to controls (Supplementary Fig.4a). At 70 days of age all niche cell frequencies are normalized to wild type levels (Supplementary Fig.4a), consistent with a normalized HSC pool (Supplementary Fig.1b). We have included the data into Supplementary Fig.4a and discussed it in the text. Please see also our response to Minor point 3.

Minor points

1. Page 8, line 233: myeloid cells should be mesenchymal cells.

Thank you, it has been corrected.

2. Page 10, line 270: Since mononuclear cells would include mesenchymal cells, mononuclear cells should be macrophages or macrophage- lineage cells.

Thank you, it has been corrected.

3. Page 11, line 306 and 322: The authors described that endothelial cells were unaltered in numbers. However, endothelial cell numbers appear to be severely reduced by histological and flow cytometric analyses (Figure 4a-c).

A potential phenotype on endothelial cells in *Rank^{cre/+};Csf1r^{fl/-}* mice was a point of concern for all three referees. The overall bone marrow cellularity in newborn *Rank^{cre/+};Csf1r^{fl/-}* mice is reduced compared to controls (Fig.1b and Fig.2c) and therefore, a fold-change approach was chosen to test whether alterations in the cellularity of all niche cell populations outnumber changes of the total bone marrow cellularity. To make this point more clear, we now have used a different way of display and have plotted frequencies of endothelial cell, MSC, and osteoprogenitors within CD45 negative bone marrow cells in Fig.4b. Using this way of display shows reduced frequencies of MSCs only. To independently test for alterations in endothelial cell numbers in *Rank^{cre/+};Csf1r^{fl/-}* mice we have performed immunohistology and also here found endothelial cell numbers comparable to wild type bones (Fig.4c). We have included this information into the text of the manuscript.

4. Page 11, line 314: Nestin is expressed in endothelial cells but not in CXCL12-expressing HSC niche cells in the mouse bone marrow (Zhou et al., Cell Stem Cell 15; 154, 2014; Baryawno et al., Cell 177; 1, 2019).

Thank you for this information. We focused on data that assessed the expression of CXCL12 in the bone marrow of neonate mice because this is the developmental phase that we analyze in our manuscript. Therefore, we cite papers that show the expression of CXCL12 in nestin-positive cells in newborn mice (Isern et al., eLife, DOI: 10.7554/eLife.03696). The expression pattern of CXCL12 in distinct niche cells differs between neonate and juvenile mice compared to adult mice (Supplementary Fig.4d). In newborn mice CXCL12 is expressed in endothelial cells, osteoprogenitors, and MSCs, whereas expression is confined to MSCs in 18 days-old mice (Supplementary Fig.4d). We have included this information into the text.

Reviewer #2 (Remarks to the Author):

The authors present a large body of work analyzing the role of yolk sac derived macrophages in sculpting the definitive HSC niche in the bone marrow. They use several conditional knock out mice and identify a number of interesting phenotypes in mice that lack embryonic but not bone marrow derived macrophages. They conclude that in early development yolk sac derived macrophages are important to regulate bone development and MSCs, with an important effect on the levels of CXCL12 production by the niche and sensing by HSCs. However, a number of conclusions remain overstated and need to be more deeply substantiated, and a number of experimental models and approaches need to be better clarified and justified. See details below.

We thank the referee for the summary, and we here provide novel data and information that we hope will clarify the questions.

Methods are referenced but missing. They must be included for the manuscript to be evaluated.

The Methods have been moved from the Supplementary Information into the manuscript.

Why are csf1r/fl used rather than fl/fl? If the rank cre is not efficient at deleting a double floxed allele, this should be shown. The mice studied are in effect already hemizygous and given the importance of embryo-derived myeloid cells, a number of the effects seen may be due to this.

We thank the referee for bringing up this point. To address this point *Rank^{cre/+};Csf1^{fl/fl}* *Rank^{cre/+};Csf1^{fl/-}*, *Csf1^{r+/-}* and *Csf1^{r+/+}* mice were generated and analyzed for leukocyte and HSPC numbers and the numbers were found the same (Figure 1, below). We conclude that there is no effect of hemizygous genotype on the phenotype. We have included this information into the text.

Figure 1: Cell counts of leukocytes (CD45+), HSPCs (KSL, Kit+ Sca-1+ Lin-), and HSCs (KSL Slam, Kit+ Sca-1+ Lin- CD48- CD150+) in the bone marrow of 3-week-old mice as indicated.

Are HSC numbers normal or normalising in csf1r rank cre mice older than the ones shown? It is important to show adult mice, especially because the higher cell cycle observed in young ones may lead to overall normal adult hematopoiesis.

We thank the referee for this insightful question. Analysis of 70-day-old mice revealed a normalized stem cell count and the data was added into Supplementary Fig.1b.

How about csf1r ko HSCs? The same assays should be shown as for the rank cre csf1r/fl mice. Should differences between the two strains be expected and are there any?

Thank you for this question. We show differences between the two strains in terms of the severity of leukocyte and KSL reduction in the bone marrow of newborn mice and we expect more differences. This may be, at least in part, due to the different genetic background of the mice which are kept on a C3H genetic background because on a C57BL/6J genetic background the frequency of knock out mice born alive is much reduced. Thus, differences between both mouse lines may be based on the different genetic background but also due to constitutive versus cell type-specific depletion of Csf1r-signaling and prompted us to focus on the in depth and mechanistic analysis of the phenotype in *Rank^{cre/+};Csf1^{fl/-}* mice. This strategy allows the exclusion of effects of Csf1r signaling on hematopoiesis from definitive HSCs.

HSC numbers in spleen and liver over time – do they balance?

HSC numbers in the spleen and liver are normalized at three weeks of age. We have included this information into to text.

Was the transcriptomic analysis also performed at 3 weeks of age? HSC Cxcr4 expression reduced in scRNAseq from adult? This finding needs to be followed up. In fact, every transcriptomics analysis needs follow up, as on its own has little value.

All mice used for the transcriptome analysis of HSCs in Fig.1 were 19-21 days of age, the information is now also included in the figure legend. The read counts for transcripts encoding for CXCR4 were found comparable between HSCs from *Rank^{cre/+};Csf1^{fl/-}* and control mice. We agree that transcriptional analysis should be followed up and we have focused on the migratory behavior of HSCs and on HSC number analysis in spleen and liver in newborn mice (both in Fig.2) because the transcriptional analysis has indicated a difference in these processes.

Why would Cxcr4 be downregulated? Could it be due to cxcl12 levels being diminished? This needs proving using the appropriate mouse crossings.

We thank the referee for this suggestion, however, we are unsure which mouse crossings would reveal the answer to this question. We have detected reduced HSC numbers in the bone marrow and spleen and increased HSC numbers in the liver of newborn mice (Fig.2c). Since CXCR4 and CXCL12 are important molecular factors involved in HSC homing we have determined their level of expression and could show that CXCR4 expression is reduced on HSCs in the bone marrow of *Rank^{cre/+};Csf1^{fl/-}* mice compared to controls (Fig.2e). This may explain the reduction in HSCs in the bone marrow at this age. While we agree that the mechanism of CXCR4 expression is interesting to investigate, we feel that this analysis goes beyond the scope of this manuscript.

Rank cre deletes from embryonic development, but is there proof that it does not delete in adult macrophages and/or osteoclasts?

We have published before that *Rank^{cre/+}* is expressed during the generation of osteoclasts (Jacome-Galarza, Percin et al., 2019 Nature, DOI: 10.1038/s41586-019-1105-7) and that osteoclasts are largely of embryonic origin that are rejuvenated by the consecutive fusion of definitive HSC-derived myeloid progenitor cells with pre-existing osteoclasts of embryonic origin (Jacome-Galarza, Percin et al., 2019 Nature, DOI: 10.1038/s41586-019-1105-7). To make this point clearer we have emphasized these findings in the text.

In Jacome-Galarza, Percin et al., we have performed lineage tracing experiments using *Rank^{cre/+};R26.eYFP^{fl/+}* mice and have detected no labeling of adult HSCs, HSPCs, monocytes, neutrophils, or eosinophils that are of adult HSC origin (Figure 2, below). Exclusively macrophage-lineage cells of embryonic origin are labeled using this lineage tracing mouse tool (Jacome-Galarza, Percin et al., 2019 Nature, DOI: 10.1038/s41586-019-

1105-7). This data is shown in Extended Data Fig 5 in the mentioned manuscript, and for clarity we provide it here for information.

[figure redacted]

Figure 2: Jacome-Galarza, Percin et al., 2019 Nature, DOI: 10.1038/s41586-019-1105-7, Extended Data Fig. 5.

Figure 3b-d work needs to be better explained and requires a deeper characterization of the mouse strains used.

We thank the referee for this suggestion and have complied.

Figure 3h-l. It is unclear how this was performed and what is the reason to present these data.

The experiments shown in figure 3h-j are described in the paragraph 'The developmental origin of macrophages impacts their functionality' in the result section. We have used *Rank^{cre/+};R26.eYFP^{f/+}* mice for this purpose and identified eYFP⁺ cells as embryonic macrophages and eYFP⁻ cells as adult macrophages. The bioinformatic analysis has revealed potential functional differences between embryonic and adult macrophages in the bone marrow. To test for such difference we have conducted the experiments shown in Fig. 3h-i and could confirm functional disparity. We have included this information into the text.

In Fig.3h we provide data assessing the presence of acidic organelles in bone marrow-derived embryonic eYFP⁺ versus adult eYFP⁻ macrophages. In the figure legend we provide information on what is shown and how it was calculated ('Fold-change of mean fluorescence intensity (MFI) of LysoTracker-positive cells within eYFP⁺ and eYFP⁻ bone marrow macrophages'). Fold-change was calculated by dividing individual MFI values of eYFP⁺ macrophages to the experimental average of the eYFP⁻ MFI values in each experiment.'. Experimental details are provided in the Material and Method section that were to be found in the Supplementary Data and that now have been moved to the manuscript.

In Fig.3i we provide data assessing the phagocytic activity of bone marrow-derived embryonic eYFP⁺ versus adult eYFP⁻ macrophages. In the figure legend we provide information on what is shown and how it was calculated ('Frequency of pHrodo positive cells within embryonic (eYFP⁺) and adult-derived (eYFP⁻) bone marrow macrophages after a 2-hours ex vivo incubation with *E.coli* bioparticles (left). Fold-change of pHrodo mean fluorescence intensity (MFI) values of eYFP⁺ and eYFP⁻ macrophages (right) Fold-change was calculated by dividing individual MFI values of eYFP⁺ macrophages to the experimental average of the eYFP⁻ MFI values in each experiment.'). Experimental details are provided in the Material and Method section that were to be found in the Extended Data and that now have been moved to the manuscript.

We have adjusted the text in the manuscript to make this point more clear.

What cells are MSCs? Nowhere it is explained what phenotypic or functional markers are used to purify these cells. Do they overlap with LepR positive cells?

Thank you for this question. We have previously published a cross-correlation of niche cell types identified by lineage-tracing tools and prospective isolation using antibody-mediated identification (Mende et al., 2019 Blood, DOI 10.1182/blood.2019000176). In this publication we have shown that LepR positive cells identified by the use of the *LepR^{cre/+}* lineage-tracer mouse tool largely contain CD45- Ter119- CD144- CD31- CD51+ Pdgfra+ Sca1- MSCs, and about 50% of MSCs identified by this cell surface phenotype are lineage-traced through the use of *LepR^{cre/+};tdRFP^{fl/wt}* mice. The tdRFP+ fraction contains both, adipogenic and osteogenic potential, whereas the tdRFP- fraction exclusively contains adipogenic potential. We have had cited this publication in the result and Material and Methods section. To make this point clearer we have added more information on the source of the identification strategy of niche cells into the text.

Fig 4 shows consequences on osteoblasts not MSCs, as CD51 expression is linked to osteoblast differentiation. Of note, CD51 is indicated as a myeloid marker in the discussion section. CD31 is greatly diminished, indicating a vascular phenotype is present too. Rewording of this results section is needed, and the findings need to be put within the context of known role of osteoclasts led bone remodeling in bone development in discussion. CD51 positive cells can be separated through *LepR^{cre/+}* lineage tracing strategies, which also allows segregation of osteogenic but not adipogenic differentiation potential (Mende et al., 2019 Blood, DOI 10.1182/blood.2019000176). The indication of CD51 as myeloid marker in the text was a mistake that has been corrected – thank you for pointing this out! A potential phenotype on endothelial cells in *Rank^{cre/+};Csf1^{fl/-}* mice was a point of concern for all three referees. The overall bone marrow cellularity in newborn *Rank^{cre/+};Csf1^{fl/-}* mice is reduced compared to controls (Fig.2c) and therefore, a fold-change approach was chosen to test whether alterations in the cellularity of all niche cell populations outnumbers changes of the total bone marrow cellularity. To make this point clearer, we now have used a different way of display and have plotted frequencies of endothelial cell, MSC, and osteoprogenitors within CD45 negative bone marrow cells in Fig.4b. Using this way of display shows reduced frequencies of MSCs only. To independently test for alterations in endothelial cell numbers in *Rank^{cre/+};Csf1^{fl/-}* mice we have performed immunohistology and also here found endothelial cell numbers comparable to wild type bones (Fig.4c). We have included this information into the text of the manuscript.

Are the ko mice smaller? Are the bones smaller/thinner/less trabecular?

Thank you for this question. *Rank^{cre/+};Csf1^{fl/-}* mice are smaller and have an osteopetrotic phenotype that we have described before (Jacome-Galarza, Percin et al., 2019 Nature, DOI: 10.1038/s41586-019-1105-7). The information was present in the introduction section but for clarity we have now included this information also into the result section of the manuscript.

Would co-culture with macrophages restore osteogenicity of MSCs?

We have conducted co-culture experiments using MSCs and embryonic or adult macrophages and detected an increase of osteogenic potential after co-culture with embryonic but not with or without co-culture with adult macrophages. The data has been included as Supplementary Fig.4b and is discussed in the revised manuscript.

Findings using prx cre are correlative not mechanistic. Are prx cre r26 YFP cells diminished in rank cre csf1r ko mice?

This question could potentially be directly addressed in *Rank^{cre/+};Csf1^{fl/-};Prx^{cre};R26.eYFP^{fl/wt}* mice, which we have not generated because it would not be possible to distinguish between Rank- or Prx-driven CRE-recombinase expression. Instead, we show in the current manuscript that the key niche cell type reduced in numbers in *Rank^{cre/+};Csf1^{fl/-}* mice are MSCs (Fig.4a-c) based on flow cytometry and histological analysis.

In Supplementary Fig.4e we provide information that *Prx^{cre}* targets the majority of MSCs and a significant proportion of OPs in newborn mice. In contrast, *Lep^{cre/+}* – an efficient marker for MSCs in adult mice (Mende et al., 2019 Blood, DOI 10.1182/blood.2019000176), and Supplementary Fig.4e) – is not expressed by neonate MSCs or OPs (Supplementary Fig.4e), which is why we chose to use *Prx^{cre}* to deplete Cxcl12 from neonate MSCs or OPs.

Discussion needs to be very clear about what age is being affected in what way. Thank you for this comment, the discussion has been revised to make sure that it is clear which age groups are discussed in each section.

Rank would affect osteoclasts so not possible to discriminate between macrophages and osteoclasts

We agree to this point and we have addressed this in the last paragraph of the discussion section and this is the key reason why we use the term ‘embryonic myeloid cells’ instead of ‘embryonic F4/80 macrophages’. However, in the current manuscript we combine mouse genetics with ex vivo and in vitro analysis and show that F4/80 embryonic macrophages are functionally different compared to F4/80 adult macrophages (Fig.3h,i), and this includes effects on MSC biology (Fig.3j and Supplementary Fig.4b) that regulate HSC migration.

Reviewer #3 (Remarks to the Author):

This is a very nice study that uses elegant mouse genetics to dissect the role of embryonic macrophages in establishment of definitive HSCs within the BM. The authors demonstrate a loss of BM HSCs in postnatal life associated with impaired fetal myeloid development through EMP-specific deletion of *Csf1r1*. The Rank-Cre system has been shown in a previously publication to definitively label and delete EMP- derived osteoclasts. The authors demonstrate that the effect on definitive HSCs is not due to an effect on HSC emergence or of deletion of *csf1r* on HSCs or other hematopoietic cells (with *Vav-Cre*), and also demonstrate that there is a compensatory increase in HSCs in the blood and spleen, suggestive of a homing defect. HSCs in mutant mice show downregulated CXCR4 and impaired homing. Finally, deletion of EMP-derived myeloid cells results in impaired MSC and osteoblast precursor development, and a reduction in CXCL12 production from MSCs, which the authors assert as a mechanisms for impaired HSC homing and seeding in the BM. The authors use additional genetic mouse models to dispel a mechanism whereby MSC-derived CXCL12 production recapitulates the phenotype of EMP-derived myeloid cell deletion. Overall, this is a very thorough study in which the data generally provide strong support for the conclusions. The findings are also quite novel and of general interest to the hematopoiesis community.

We are thrilled by the positive comments of this referee and would like to point out that we felt that this is an excellent summary of the focus and rationale of the experiments provided in the manuscript.

Nonetheless, there are a few specific areas where the manuscript could potentially be improved:

In general, the authors use their Rank-Cre X *Csf1r1*^{Fl/Fl} model to delete “fetal-derived myeloid cells” in the bone marrow. Not until the very end of the manuscript do they mention osteoclasts. The reason for this omission is unclear to the reviewer. I think it would be beneficial to the manuscript to state/address this earlier, perhaps when discussing the identification of the F4/80 population in Fig. 3.

Thank you for this suggestion! We would like to point out that we have provided information on the embryonic origin of osteoclasts in the introduction (‘In the bone marrow, only highly specialized macrophages, the osteoclasts, are so far considered to be of embryonic origin’). Further, in the discussion we highlight that we formally cannot distinguish between the contribution of embryo-derived macrophages or osteoclasts in their contribution to the establishment of the bone marrow niche (‘While we formally cannot distinguish between the contribution of embryo-derived bone marrow resident osteoclasts and macrophages to the

establishment of the niche, we use genetic tools and provide functional evidence *ex vivo* to show that embryonic myeloid cells orchestrate the non-hematopoietic niche space important for the normal founding of the hematopoietic stem cell pool.’). However, we considered it a great idea to improve clarity on this point by additional inclusion of this information in the discussion of the identification of the F4/80 population in Fig.3, which we have carried out.

In Figure 1F, the authors state that they transplant BM HSC (line 132), but they are actually transplanting LSK. These are not the same, and there is a difference in the frequency of HSCs within LSK between WT and mutant. This should be justified and/or addressed within the results and discussion.

Given this difference, the differences in outcomes between 1F and 2F are somewhat confusing. In 1F, mutant LSK should have less HSCs, but they appear to have more HSCs in the BM at 32 weeks. Why would transfer of WBM in 2F yield such different results as compared to LSK?

We agree that this is an important point. To improve clarity we have rephrased ‘HSCs’ in line 132 to ‘stem and progenitor cells (KSL)’.

Fig.1f: The experiment showed no difference in donor cell contribution between KSL from *Rank^{cre/+};Csf1^{fl/-}* or control mice (3 weeks of age) despite a reduced overall frequency of KSL (HSPC) and KSL Slam (HSC) in CD45+ bone marrow cells. Absolute numbers of HSPCs as well as HSCs are reduced in *Rank^{cre/+};Csf1^{fl/-}* mice, and the reduction is comparable in the fold-change analysis depicted in Fig.1b. Thus, when transplanting HSPCs the frequency (and absolute numbers) of HSCs within this cell compartment is the same in *Rank^{cre/+};Csf1^{fl/-}* and control mice. This is why this experiment shows that on a per-cell basis the function of HSCs is not altered – there are ‘just’ fewer of them in the bone marrow.

Fig.2f: In this experiment total bone marrow cells from 2 days-old mice were transplanted, thus, in this experimental setting, stem and progenitor cells are significantly decreased in the donor cells from *Rank^{cre/+};Csf1^{fl/-}* compared to control mice (Fig.2c). This explains why the repopulation data shown in Fig.1f and 2f differ.

We consider this an important point and have included donor cell information into Fig.2f and adjusted the text passages.

In Fig 2H – Did cells that did not go to the BM home to spleen and PB instead?

Unfortunately, we have not analyzed donor cells in the spleen or blood in this experiment and therefore cannot answer this question.

In Figure 3, the RNA seq data would benefit from improved data presentation and some further analysis. Data in SFig3 are a bit confusing, particularly SFig 3E. It would be beneficial if the authors could identify some of the critical downregulated genes associated with myeloid leukocyte migration that they think are drivers of the observed effects.

Thank you for the helpful suggestion. In response, we have now included the GSEA analysis into Fig.3g and replaced the semantic space graph with heatmaps displaying expression levels of genes included in the significantly enriched GO terms *myeloid leukocyte migration* and *myeloid leukocyte activation* in Supplementary Fig.3e. We have also added details about specific genes deregulated in embryonic macrophages associated with myeloid leukocyte migration and activation in the text. Finally, we have analyzed the enrichment of biological processes in genes differentially expressed between eYFP+ embryonic- and eYFP- adult-derived bone marrow macrophages and included the results as Supplementary Fig.3f. We have discussed all findings in the text.

Given that embryonic macrophages support MSCs, is there any information in the RNA-seq dataset to also support this *in vivo* observation?

We thank the reviewer for this question. To determine specific molecular interactions potentially mediating differential effects of embryonic versus adult macrophages on MSC function we extracted receptor-ligand pairs using a previously published database (<https://github.com/hoefler-lab/CellInteractionScores>). The results of this analysis suggest that macrophages of distinct developmental origin differ in their effect on extracellular matrix

organization. The results of this analysis are shown in Supplementary Fig.4c and are discussed in the text.

Figure 3J – presentation of data could be simplified by utilizing label on the X axis. We have complied to this request and now find it easier to understand.

In Figure 4; the authors describe “Unchanged endothelial cells” – line 233 – but 4a appears to be considerably less ECs, and the conversion for “fold change” in 4b is a bit confusing and not intuitive. Can the authors provide more explanation?

A potential phenotype on endothelial cells in *Rank^{cre/+};Csf1^{fl/-}* mice was a point of concern for all three referees. The overall bone marrow cellularity in newborn *Rank^{cre/+};Csf1^{fl/-}* mice is reduced compared to controls (Fig.1b and Fig.2c) and therefore, a fold-change approach was chosen to test whether alterations in the cellularity of all niche cell populations outnumbers changes of the total bone marrow cellularity. To make this point more clear, we now have used a different way of display and have plotted frequencies of endothelial cell, MSC, and osteoprogenitors within CD45 negative bone marrow cells in Fig.4b. Using this way of display shows reduced frequencies of MSCs only. To independently test for alterations in endothelial cell numbers in *Rank^{cre/+};Csf1^{fl/-}* mice we have performed immunohistology and also here found endothelial cell numbers comparable to wild type bones (Fig.4c). We have included this information into the text of the manuscript.

Is HSC function similarly unperturbed when CXCL12 is deleted from MSCs?

We appreciate the referee’s question. Consistent with earlier studies using adult *Prx^{cre/+};CXCL12^{fl/-}* or *Prx^{cre/+};CXCL12^{fl/fl}* mice (Greenbaum et al., 2013, *Nature*; Ding et al., 2013, *Nature*, respectively), we observed a notable reduction in HSC numbers within the bone marrow of newborn *Prx^{cre/+};CXCL12^{fl/fl}* mice (Fig.4). Further, Greenbaum et al. reported increased HSPC potential in blood and spleen, recapitulating the phenotype we observe in 3 week old *Rank^{cre/+};Csf1^{fl/-}* mice, highlighting functional similarities between HSCs in these two models. Finally, Greenbaum et al. have performed competitive transplantation experiments that revealed decreased numbers of HSCs, however donor HSCs from *Prx^{cre/+};CXCL12^{fl/-}* mice contained full potential to contribute to hematopoiesis in the recipient mice, suggesting unperturbed function of HSCs. We have included this information into the discussion section.

Point-by-point response to the reviewer's comments, MS NCOMMS-24-20946-T

Reviewer #1 (Remarks to the Author):

The authors have given a satisfactory response to my concerns, improving the manuscript with successfully. Since the additional data showing that the frequency of common lymphoid progenitors (CLPs) was reduced in the bone marrow of Rank-cre; Csf1r fl/- mice at the age of 3 weeks is interesting, I would recommend the authors to show it in the main Figure and not in the Supplementary Figure 1a.

We thank the referee for this suggestion and agree that this finding is interesting, however, we decided to keep the information in the supplementary information since it is not directly relevant for the point that we address in the manuscript.

Reviewer #2 (Remarks to the Author):

I commend the authors for their thorough revisions, for highlighting how the present manuscript relates to and builds on their prior studies, and for adding excellent clarity to many points. All my concerns have been addressed.

We thank the referee these kind words, we agree that the manuscript was much improved through this review process.

Reviewer #3 (Remarks to the Author):

In this reviewed manuscript, the authors have satisfactorily addressed all previous concerns and thereby significantly improved the manuscript.

We thank the referee for this statement and are excited to move forward with publication.